# Ancient human DNA recovered from a Palaeolithic pendant

Elena Essel[1,17 ✉], Elena I. Zavala[1,2,13], Ellen Schulz-Kornas[3,17], Maxim B. Kozlikin[4], Helen Fewlass[1], Benjamin Vernot[1], Michael V. Shunkov[4], Anatoly P. Derevianko[4], Katerina Douka[5,6], Ian Barnes[7], Marie-Cécile Soulier[8], Anna Schmidt[1], Merlin Szymanski[1], Tsenka Tsanova[1,14], Nikolay Sirakov[9], Elena Endarova[10], Shannon P. McPherron[1], Jean-Jacques Hublin[1,15], Janet Kelso[1], Svante Pääbo[1], Mateja Hajdinjak[11,16], Marie Soressi[12,17 ✉] & Matthias Meyer[1,17 ✉]

Artefacts made from stones, bones and teeth are fundamental to our understanding of human subsistence strategies, behaviour and culture in the Pleistocene. Although these resources are plentiful, it is impossible to associate artefacts to specific human individuals[1] who can be morphologically or genetically characterized, unless they are found within burials, which are rare in this time period. Thus, our ability to discern the societal roles of Pleistocene individuals based on their biological sex or genetic ancestry is limited[2–5]. Here we report the development of a non-destructive method for the gradual release of DNA trapped in ancient bone and tooth artefacts. Application of the method to an Upper Palaeolithic deer tooth pendant from Denisova Cave, Russia, resulted in the recovery of ancient human and deer mitochondrial genomes, which allowed us to estimate the age of the pendant at approximately 19,000–25,000 years. Nuclear DNA analysis identifies the presumed maker or wearer of the pendant as a female individual with strong genetic affinities to a group of Ancient North Eurasian individuals who lived around the same time but were previously found only further east in Siberia. Our work redefines how cultural and genetic records can be linked in prehistoric archaeology.

Palaeolithic assemblages typically contain a multitude of objects that may differ in age by hundreds or thousands of years, even when found in close proximity[1]. Thus, it can be challenging to associate human remains with specific objects. Recent advances in the retrieval of human DNA from sediments[6–8] can be used to connect artefacts with genetic populations. However, precise identification of the specific makers or users of these objects would require the recovery of human DNA directly from the objects themselves, analogous to modern-day forensic investigations. In theory, such analyses are most promising for artefacts made from animal bones or teeth, not only because they are porous and thereby conducive to the penetration of body fluids (for example, sweat, blood or saliva) but also because they contain hydroxyapatite, which is known to adsorb DNA and reduce its degradation by hydrolysis and nuclease activity[9,10]. Ancient bones and teeth may therefore function as a trap not only for DNA that is released within an organism during its lifetime and subsequent decomposition but also for exogenous DNA that enters the matrix post-mortem through microbial colonization[11] or handling by humans. However, DNA extraction from ancient skeletal material either requires destructive sampling, or risks alteration of specimens if they are directly submerged in extraction buffer[12,13]. Conservation is a primary concern because of the scarcity of bone and tooth artefacts at Pleistocene sites, especially of pendants and other ornaments that were extensively handled or worn in close body contact. We therefore set out to develop a method for DNA isolation from bones and teeth that preserves the integrity of the material, including surface microtopography, and to investigate the possibility of DNA retrieval from bone and tooth artefacts.

## A non-destructive DNA isolation method

To identify reagents compatible with non-destructive DNA extraction, we selected ten unmodified faunal remains from the Palaeolithic sites of Quinçay and Les Cottés in France (Extended Data Table 1 and Supplementary Information 1), which were similar in size and shape

[1]Max Planck Institute for Evolutionary Anthropology, Leipzig, Germany. [2]Department of Biology, San Francisco State University, San Francisco, CA, USA. [3]Department of Cariology, Endodontology and Periodontology, University of Leipzig, Leipzig, Germany. [4]Institute of Archaeology and Ethnography, Siberian Branch, Russian Academy of Sciences, Novosibirsk, Russia. [5]Department of Evolutionary Anthropology, Faculty of Life Sciences, University of Vienna, Vienna, Austria. [6]Human Evolution and Archaeological Sciences (HEAS) Research Network, University of Vienna, Vienna, Austria. [7]Earth Sciences Department, Natural History Museum, London, UK. [8]Maison de la Recherche, Université de Toulouse-Jean Jaurès, CNRS UMR 5608 TRACES, Toulouse, France. [9]National Institute of Archaeology with Museum, Bulgarian Academy of Sciences, Sofia, Bulgaria. [10]National Museum of History, Sofia, Bulgaria. [11]Ancient Genomics Laboratory, The Francis Crick Institute, London, UK. [12]Faculty of Archaeology, Leiden University, Leiden, The Netherlands. [13]Present address: Department of Molecular and Cell Biology, University of California, Berkeley, CA, USA. [14]Present address: Department of Chemistry "Giacomo Ciamician", Alma Mater Studiorum, University of Bologna, Bologna, Italy. [15]Present address: Chaire de Paléoanthropologie, Collège de France, Paris, France. [16]Present address: Max Planck Institute for Evolutionary Anthropology, Leipzig, Germany. [17]These authors contributed equally: Elena Essel, Elena I. Zavala, Ellen Schulz-Kornas, Marie Soressi, Matthias Meyer. ✉e-mail: elena_essel@eva.mpg.de; m.a.soressi@arch.leidenuniv.nl; mmeyer@eva.mpg.de

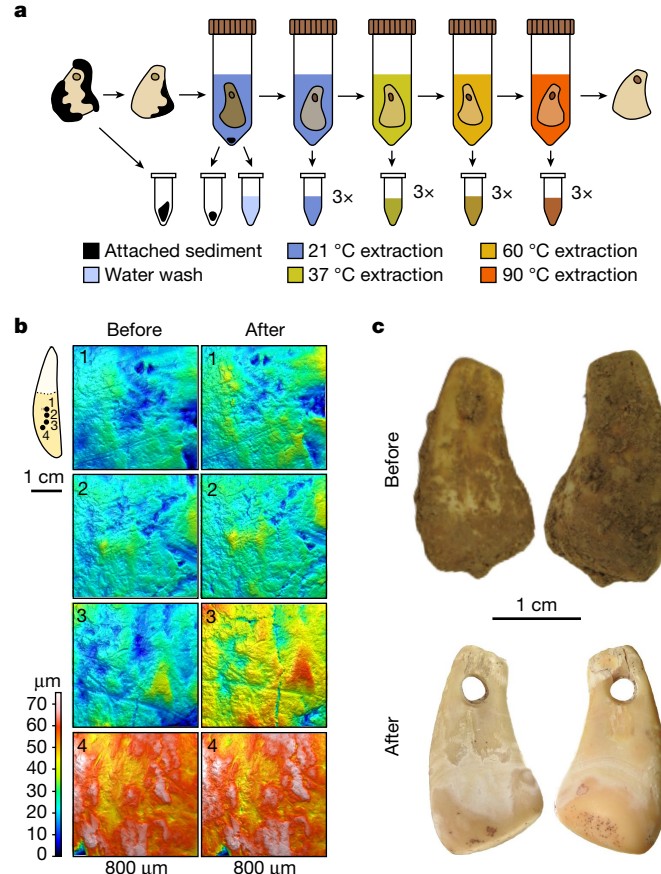

**Fig. 1 | Overview of the non-destructive DNA extraction method.**
**a**, Workflow of the gradual, non-destructive DNA extraction method using sodium phosphate buffer at elevated temperatures. **b**, Four 3D surface texture measurements (1–4) indicated on the outline of a tooth used for testing (SP6649) before and after non-destructive DNA extraction showing no substantial surface alterations. **c**, Photographs of DCP1 before and after cleaning and non-destructive DNA extraction.

to material typically used for osseous artefact production, and submerged them in several reagents previously used in ancient DNA extraction, as well as in water for comparison. These included (1) a guanidinium thiocyanate-containing reagent previously suggested for non-destructive DNA extraction[12], (2) an ethylenediaminetetraacetate (EDTA) solution, which is a decalcifier commonly used in ancient DNA extraction[14–16], (3) a sodium hypochlorite (bleach) solution, which is an oxidizing reagent used to remove surface-exposed contaminant DNA[11,17], and (4) a sodium phosphate buffer supplemented with detergent[11], which has been recently shown to enable temperature-controlled DNA release from powdered bone samples[18].

Mapping of the microtopography using quantitative 3D surface texture analysis[19,20] before and after the treatments revealed substantial surface alterations on all objects exposed to either the guanidinium thiocyanate reagent or EDTA (Extended Data Fig. 1 and Supplementary Information 2). By contrast, only sporadic and smaller alterations were detected with the other reagents, including sodium phosphate buffer (Fig. 1b), possibly due to the removal of traces of sediment and other small particles, as indicated by visible changes in coloration of some of the objects (Extended Data Fig. 2). On the basis of these results, we developed a non-destructive DNA isolation method for the stepwise release of DNA from the bone or tooth matrix using serial incubations in sodium phosphate buffer at 21, 37, 60 and 90 °C, with three incubations per temperature (Fig. 1a).

We then applied this method to 11 osseous objects, labelled Q10 to Q19 as well as Q27, that were excavated several decades ago in the Châtelperronian layers of Quinçay Cave in France and that had potentially been used as tools some 35–45 thousand years ago (ka)[21] (Extended Data Table 1 and Extended Data Fig. 3). We prepared single-stranded DNA libraries[22,23] from the first DNA fraction recovered at each temperature, and enriched the libraries for mammalian mitochondrial (mt) DNA[24]. A metagenomic pipeline for assigning sequenced mtDNA fragments to mammalian taxa on the biological family level[6] identified 1,628 cervid mtDNA fragments in the 60 °C and 90 °C fractions of object Q10, a reindeer bone (109 and 1,519 fragments, respectively; Fig. 2 and Supplementary Data 1). These fragments showed elevated frequencies of cytosine (C)-to-thymine (T) substitutions at their ends, consistent with deamination of cytosine residues seen in ancient DNA[6,25] (Supplementary Data 1). Another object, Q15, which was made of ivory, yielded 2,004 elephantid mtDNA sequences with elevated frequencies of C-to-T substitutions in the 37 °C, 60 °C and 90 °C fractions (248, 325 and 1,431 fragments, respectively). In addition, we identified hominid and suid mtDNA fragments with no evidence for ancient DNA base damage in every DNA fraction from the 11 objects, thus resulting from contamination with human and pig DNA after excavation. Human DNA contamination was particularly severe, amounting to between 70.9% and 98.3% of the identified mtDNA fragments (between 293 and 92,949 fragments in total; 17,627 on average), thereby potentially masking traces of ancient human or other mammalian DNA.

## Studying freshly excavated artefacts

As present-day human DNA contamination seemed to be ubiquitous on surfaces of objects that were handled with bare hands during and after excavation, we collected artefacts from ongoing excavations at two Palaeolithic sites, using gloves and facemasks as soon as they became partly exposed to prevent contamination. At Bacho Kiro Cave in Bulgaria, we recovered three Upper Palaeolithic tooth pendants (henceforth 'BKP1–BKP3') from layers I, H/I and I/J of niche 1 (Extended Data Fig. 3). At Denisova Cave, a tooth pendant ('DCP1') was recovered from layer 11 of the south chamber (Fig. 1c and Extended Data Fig. 4).

Larger clumps of sediment adhering to the artefacts were manually removed, and the artefacts were subsequently cleaned by three successive water washes. DNA was then extracted from the sediment clumps, the water was used for washing and the sediment particles were collected in this process ('sediment pellets'), as well as directly from the artefacts using the non-destructive method described above. Ancient mammalian mtDNA was detected in all fractions that were analysed, except for some of the water washes and the associated sediment pellets (Fig. 2). The trajectories of DNA released from the four artefacts were similar in that the highest yield of mammalian mtDNA was obtained at 90 °C in the phosphate-based DNA extraction (up to 734, 6,614, 456 and 77,910 mtDNA fragments for BKP1, BKP2, BKP3 and DCP1, respectively; Supplementary Data 1). However, library preparation efficiencies were low for the Bacho Kiro Cave material (less than 10% for many fractions), presumably due to the co-extraction of inhibitory substances (Fig. 2), indicating that more DNA was released than could be recovered and sequenced.

The phosphate DNA fractions obtained at 37 °C, 60 °C and 90 °C are dominated by ancient ursid mtDNA fragments for BKP2 and BKP3, and cervid mtDNA fragments for DCP1, in agreement with their morphological identification (Extended Data Table 1). For BKP1, which is morphologically indeterminate, the phosphate fractions are dominated by bovid mtDNA fragments. In contrast to the phosphate fractions, DNA recovered from the sediment adhering to the artefacts is taxonomically more heterogeneous (Fig. 2). In addition, substantially lower numbers of human mtDNA fragments (between 0 and 2,969 per fraction, 246 on average; Supplementary Data 1) were recovered from the freshly excavated artefacts than from the Quinçay material. Similarly, very

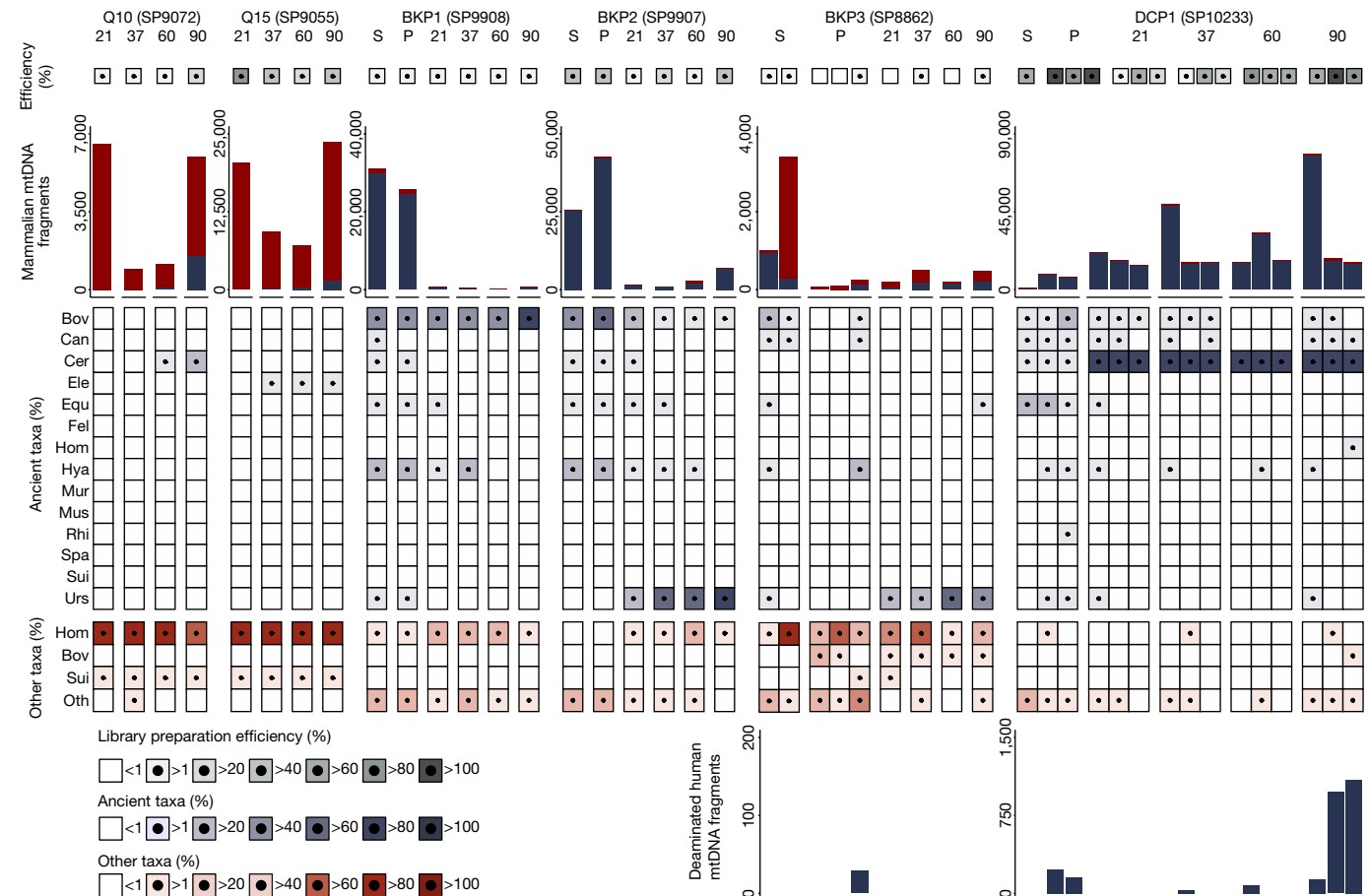

**Fig. 2 | Number and taxonomic composition of mammalian and human mtDNA fragments recovered from six artefacts during stepwise, non-destructive DNA extraction.** DNA fractions are denoted S (attached sediment), P (sediment pellet recovered during water wash), and 21, 37, 60 and 90 (three incubations in phosphate buffer at the indicated temperatures in °C). Low library preparation efficiencies indicate reduced DNA recovery due to the co-extraction of inhibitory substances. Assignments to 'ancient' and 'other'

taxa were performed independently for each family based on the significance of evidence for cytosine deamination. The bottom right chart shows the number of deaminated hominin mtDNA fragments recovered in each fraction after hominin-specific mtDNA capture (only positive fractions). Bov, Bovidae; Can, Canidae; Cer, Cervidae; Ele, Elephantidae; Equ, Equidae; Fel, Felidae; Hom, Hominidae; Hya, Hyaenidae; Oth, other; Mur, Muridae; Mus, Mustelidae; Rhi, Rhinocerotidae; Spa, Spalacidae; Sui, Suidae; Urs, Ursidae.

few suid mtDNA fragments (15 or less) were recovered, indicating that little post-excavation contamination had occurred. Of note, significant signals of cytosine deamination were observed among the human mtDNA fragments recovered in one of the 90 °C fractions from DCP1.

To increase human DNA recovery, we enriched all libraries again using a capture probe set specifically targeting human mtDNA. For the Bacho Kiro Cave material, this enabled the detection of small traces of ancient human DNA in a sediment pellet from BKP3 (29 deaminated mtDNA fragments), but none of the other fractions. For DCP1, human mtDNA fragments with significant evidence for ancient DNA base damage were identified in the first two sediment pellets recovered from the water washes, the first 37 °C and 60 °C fractions, and all three 90 °C fractions (Fig. 2). The largest numbers of deaminated human mtDNA fragments were obtained in the second and third 90 °C fractions (971 and 1,096, respectively), indicating that extended incubation at high temperature enabled ancient human DNA release from the pendant.

Preparation of additional libraries from the second 90 °C fraction of DCP1, the fraction with the lowest estimate of present-day human contamination (0.1%, 95% CI: 0.0–2.8%), yielded 62-fold average coverage of the human mtDNA genome and a near-complete consensus sequence (Supplementary Information 5). This sequence, which falls together with mtDNA sequences assigned to haplogroup U in a phylogenetic tree (Fig. 3a), contains seven 'diagnostic' positions that distinguish it from the mtDNA sequences of other human individuals (Extended Data

Table 2). Among the mtDNA fragments overlapping these positions, 86.6% (95% CI: 82.2–90.5%) match the state of DCP1 (Extended Data Table 3), suggesting that the mtDNA fragments recovered in this fraction originate predominantly, but not exclusively, from a single ancient human individual, presumably the user or the maker of the pendant. Support for the DCP1 consensus sequence is slightly lower in the first (77.8%, 95% CI: 40.0–97.2%) and third (82.8%, 95% CI: 78.9–86.2%) 90 °C phosphate fractions, consistent with the slightly higher estimates of present-day human contamination in these fractions (12.8%, 95% CI: 1.0–24.6% and 6.6%, 95% CI: 4.3–8.9%, respectively). By contrast, support for the DCP1 consensus is low in the preceding 60 °C fraction (37.5%, 95% CI: 8.5–75.5%), indeterminate in the 37 °C fraction and low in the first (20.0%, 95% CI: 5.7–43.7%) and second (9.5%, 95% CI: 1.2–30.4%) sediment pellets. These results indicate that the initial water washes and incubations in phosphate buffer at below 90 °C primarily released ancient human DNA from one or more other individuals, which were present in smaller quantities in the surrounding sediment or had adsorbed directly to the surface of DCP1.

## Phylogenetic analyses and dating
On the basis of the branch length of the DCP1 consensus sequence in a tree with other present-day and ancient human mtDNA genomes (Fig. 3a), we estimated its age to 18.5 thousand years (kyr), with a 95%

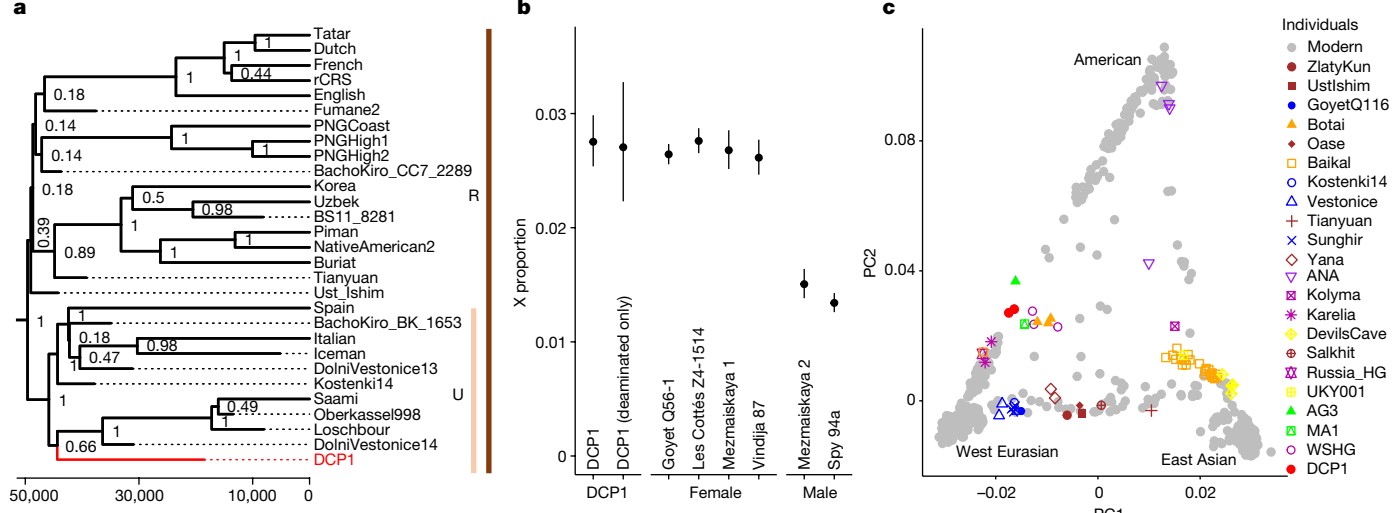

**Fig. 3 | Ancient human mtDNA and nuclear DNA isolated from DCP1.**
**a**, The position of DCP1 in a Bayesian tree reconstructed from modern[31,32] and ancient[33] human mtDNA sequences (see Supplementary Information 5 for the full tree). Nodes are labelled with the corresponding posterior probabilities, and the *x* axis represents years from the present. Identified haplogroups are outlined by the bars on the right. rCRS, revised Cambridge reference sequence. **b**, X–autosome proportion in DCP1 (using all and deaminated molecules only) in comparison to data from six other ancient hominin individuals[34,35]. Circles correspond to the calculated values of the ratios for the number of X to

(X + autosomal) fragments for each individual (*n* (of single-nucleotide polymorphisms (SNPs)) = 20,526, 3,734, 124,862, 85,901, 34,756, 41,632, 34,677 and 72,992 SNPs for each calculation, as ordered on the *x* axis). The error bars represent 95% binomial CIs of the measurement in each individual. **c**, Principal component (PC) analysis of non-African modern human genomes[36] (grey) with ancient human genomes (coloured) projected on top. DCP1 was analysed twice, using all data versus deaminated fragments only. AG3, Afontova Gora 3; ANA, ancient Native Americans; MA1, Mal'ta 1; Russia_HG, Russian hunter–gatherers; UKY001, Ust Kyakhta; WSHG, West Siberian hunter–gatherers.

highest posterior density interval ranging from 4.6 to 31.6 kyr (Supplementary Information 5). Furthermore, we used cervid mtDNA probes to reconstruct the complete mtDNA genome of the tooth, identified as wapiti (*Cervus canadensis*), at 635-fold coverage. A tree with eight additional ancient wapiti mtDNA genomes of known age generated in this study (Supplementary Information 6), estimates the age of DCP1 at 24.7 kyr (highest posterior density interval of 12.8–39.0 kyr). Both genetic ages are consistent with each other and with the younger of two radiocarbon dates that we obtained from charcoal discovered in the proximity of DCP1 in layer 11 (OxA-X-3089-11: 24,200–23,830 calibrated years before present (cal bp) and OxA-X-3089-12: 39,180–37,560 cal bp) at 95.4% probability (Supplementary Information 1). We therefore suggest that genetic dating obviates the need for direct radiocarbon dating of the pendant, although this remains technically possible after non-destructive DNA extraction (Supplementary Information 3).

For nuclear DNA analysis, hybridization capture was performed using libraries from the second and third 90 °C phosphate fractions, targeting sites in the human genome that are known to be polymorphic in modern or archaic humans and that are located in regions of high sequence divergence between humans and other mammals[8]. Sequence information was obtained for 336,429 of these sites (71.5% of the sites targeted), with estimates of present-day human and faunal contamination both below 1%. Comparisons with present-day human populations[26] using *f*3-statistics and *D*-statistics[27,28] show high affinities to Native Americans (Extended Data Fig. 5). When projected into a principal component analysis with other ancient human individuals (Fig. 3c), DCP1 falls within a group of Ancient North Eurasian individuals from further east in Siberia, which includes the approximately 24 ka Mal'ta 1 and the approximately 17 ka Afontova Gora 3 individuals[29,30]. Both of these individuals are genetically closer to DCP1 than non-Ancient North Eurasian individuals when tested with *D*-statistics (Extended Data Fig. 6b), and all three show similar affinities to ancient Siberians and Native Americans with *f*3-statistics and *D*-statistics (Extended Data Fig. 6a,c). In addition, shotgun data were produced from one of the libraries to allow a comparison of sequence coverage for the X

chromosome and the autosomes, which is compatible with the human DNA in the 90 °C fraction originating predominantly from a female individual (Fig. 3b and Supplementary Information 7).

## Conclusions

In summary, our work highlights that artefacts made from bones or teeth are a previously untapped source of ancient human DNA that can provide insights about the ancestry and biological sex of the individuals who handled, carried or wore these objects in the deep past. The non-destructive DNA extraction method reported here allows a stepwise release of this DNA, making it possible to distinguish DNA that penetrated deeply into an object during its manufacture or use from DNA that may originate from the surrounding sediment. Of note, the coverage depth of targeted sites in the human nuclear genome achieved from DCP1 is similar to what has been obtained with hybridization capture from well-preserved Pleistocene human remains[30]. Furthermore, the recovery of both human and faunal DNA enabled two independent genetic estimates of its age.

Further work is needed to determine how often human DNA can be recovered from Palaeolithic osseous artefacts. As surface DNA contamination can hamper these analyses, we urge archaeologists to apply protocols for minimizing handling during and after excavation. If this is done, it might become possible to systematically combine genetic and cultural analyses to study Pleistocene artefact use and uncover possible task specialization by individuals of a particular biological sex or genetic ancestry.

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

# Methods

## Sample collection

For reagent testing, we collected ten unmodified faunal remains, similar in and size and shape to material used in artefact production, from the late Middle Palaeolithic and early Upper Palaeolithic deposits of the French sites Quinçay (seven) and Les Cottés (three) (Extended Data Table 1 and Extended Data Fig. 2). The estimated ages of these specimens range from 55 to 35 kyr[21,37,38]. Non-destructive DNA extraction was then applied to 15 osseous specimens excavated at Quinçay Cave, all from layers attributed to the Châtelperronian technocomplex and probably dating to 45–35 ka[21,37], to three tooth pendants excavated in the Initial Upper Palaeolithic (45–43 ka)[39] layers in the niche 1 area of Bacho Kiro Cave[40,41], as well as on one tooth pendant excavated in 2019 in layer 11 (39–24 ka), square E-3, of the south chamber of Denisova Cave (Extended Data Table 1 and Extended Data Fig. 3). Samples from Bacho Kiro and Denisova Cave were excavated and handled using sterile gloves and additional precautions were taken to minimize the introduction of modern DNA contamination. More information about the samples and the archaeological context in which they were recovered is provided in Supplementary Information 1.

## Testing reagents for non-destructive DNA extraction using 3D surface texture measurements

We evaluated four reagents for their potential use in non-destructive DNA extraction by applying each to two Pleistocene samples comparable with the ones usually transformed into bone or tooth tools and ornaments, preferably choosing one bone and one tooth fragment for each reagent (Extended Data Table 1). As none of the objects was perfectly clean and sediment microparticles may be released upon exposure to liquids, we also performed treatments with water to obtain baseline measurements of changes in the microtopography that are independent of the chemical compositions of the reagents. Reagents and incubation conditions were as follows:

(1) Following a protocol[42] further detailing the non-destructive DNA extraction method for museum specimens by Rohland et al. (2004)[12], samples were completely submerged in 10–15 ml guanidinium thiocyanate buffer (5 M guanidinium isothiocyanate, 50 mM Tris-HCl, pH 8.0, 25 mM NaCl, 1.3% Triton X-100, 20 mM EDTA, pH 8.0, and 50 mM DTT) and incubated for 5 days in the dark.

(2) Incubation in EDTA, supplemented with a detergent (0.45 M EDTA, pH 8.0, and 0.05% Tween-20), was performed for 15 min at room temperature. EDTA is widely used in bone extraction protocols as a decalcifying agent to dissolve hydroxyapatite, the main mineral component of bones[14–16]. It is therefore expected to cause severe alterations to the sample material and was included to demonstrate the effect of such alterations on the quantitative 3D-surface texture analysis (3DST) measurements.

(3) Two samples were submerged for 15 min in 0.5% sodium hypochlorite (bleach) solution, a reagent previously used for decontamination of ancient skeletal remains[11,17]. Bleach treatment destroys surface-bound DNA and is not suitable for DNA extraction. However, it may be used in later implementations of the method to remove contaminant DNA before non-destructive DNA extraction.

(4) Temperature-controlled release of DNA was performed following a modified version of the method by Essel et al. (2021)[18] by submerging the samples in sodium phosphate buffer (0.5 M sodium phosphate, pH 7.0, and 0.1% Tween-20). Serial incubations were performed at 21 (room temperature), 37, 60 and 90 °C, each with incubation times of 30 min for a total of three incubations per temperature.

All treatments were performed in 50-ml Falcon tubes without agitation to avoid any mechanical damage to the sample. Reagent volumes were chosen individually for each sample (ranging from 5 ml to 17.5 ml) to ensure complete submergence. Incubations above room temperature

were performed in a Heating-ThermoMixer MHR 11 (Hettich Benelux) equipped with inserts for 50-ml Falcon tubes. For the 90 °C incubation in phosphate buffer, the temperature of the device was set to 99 °C to ensure that 90 °C was reached inside the tube by the end of the incubation time. After the treatments, all samples were placed in fresh 50-ml Falcon tubes and incubated in water for 1 h at room temperature to remove residual reagents. Samples were dried at room temperature for 5 days before they were returned to their storage containers.

Changes in the microtopography of the bone or tooth objects were tracked using quantitative 3DST[19] before and after extraction following established protocols[19,43]. Meshed axiomatic 3D models were generated and the following ISO 25178 parameters were used for statistical testing (paired Student's $t$-test, before and after the treatment, $\alpha \leq 0.01$): mean roughness ($Sa$), void volume ($Vvv$), peak curvature ($Spc$) and peak density ($Spd$). Further details regarding the 3DST measurements are provided in Supplementary Information 2.

In addition, we explored the compatibility of phosphate-based non-destructive DNA extraction with subsequent [14]C dating (Supplementary Information 3).

## Non-destructive DNA isolation from artefacts

As sodium phosphate buffer did not cause substantial alterations of ancient bones and teeth in 3DST measurements, we modified a previously described method for temperature-controlled gradual DNA release from bone and tooth powder using this reagent[18] for non-destructive DNA extraction from complete bones and teeth. Stepwise extraction of DNA makes it possible to closely monitor the release of different DNA components during the extraction process (endogenous DNA, environmental DNA from the surrounding sediment, ancient human DNA and present-day contamination), potentially allowing inferences to be drawn about whether these components originate from traces of sediment that may still be adherent to the object, from its surface or its interior. We then applied this protocol to a total of 15 specimens from Quinçay, Bacho Kiro Cave and Denisova Cave. Temperature-controlled non-destructive DNA extraction was performed in an ancient DNA clean-room at the Max Planck Institute for Evolutionary Anthropology in Leipzig, Germany, according to the four steps below (see Fig. 1a for a schematic overview):

**(1) Removal of sediment.** This step was only performed for the freshly excavated material. First, clumps of sediment attached to the specimen were carefully removed by hand using a flexible disposable plastic microspatula. The specimen was then put into a 50-ml Falcon tube, rinsed by pouring between 20 ml and 50 ml of water into the tube, and transferred to a fresh Falcon tube. This procedure was repeated two to three times until no more sediment was released into the water. The tubes with water containing the sediment that had been washed off were then centrifuged at 16,400$g$ for 5 min to pellet the sediment, and the clear supernatants were transferred to fresh tubes. All three types of material collected in this procedure (the manually removed sediment, the sediment pellet collected by centrifugation and the clear water) were subsequently subjected to DNA purification (see below).

**(2) Temperature-controlled DNA release using sodium phosphate buffer.** Each cleaned specimen was put into a 50-ml Falcon tube to which sodium phosphate buffer (0.5 M sodium phosphate, pH 7.0, and 0.1% Tween-20) was added until the specimen was completely submerged in the reagent (between 5 ml and 50 ml). After 30 min of incubation at room temperature (without agitation), the buffer was transferred to a fresh 50-ml Falcon tube. This step was repeated twice at room temperature (21 °C, for a total of three incubations) and then three times each at 37 °C, 60 °C and 90 °C (see above for details on the device and temperature settings used). A final incubation in water was performed at room temperature to remove residual reagent and the specimens were dried at room temperature for 5 days.

**(3) DNA concentration.** To facilitate subsequent DNA purification, half of the volumes of the water and phosphate buffer DNA fractions generated in steps (1) and (2) were reduced to between 50 µl and 75 µl by concentrating the DNA using Amicon Ultra-4 Centrifugal Filter Units with Ultracel-3 membranes (Millipore). For this, up to 4 ml or 15 ml of the respective sample was added to a filter unit, which was spun for 90 min at 4,000$g$ in a centrifuge with an active cooling unit set to 21 °C. In cases in which the sample volume exceeded 4 ml or 15 ml, the flow-through was discarded and the filter unit was reloaded with remaining sample. Finally, a buffer exchange was performed by adding 4 ml or 15 ml TE buffer (10 mM Tris-HCl, pH 8.0, and 1 mM EDTA) to the concentrated sample on top of the filter unit and spinning for 30 min at 4,000$g$. The supernatant was filled up to 300 µl with TE buffer (10 mM Tris-HCl, 1 mM EDTA, pH 8.0) and then transferred into a fresh 1.5-ml Eppendorf low-bind tube and stored at −20 °C until further processing.

**(4) DNA purification.** DNA was isolated from the concentrated DNA fractions prepared in step (3) using a column-based method for silica-based ancient DNA extraction detailed elsewhere[44]. For this, 300 µl concentrated DNA was used as input for DNA purification using binding buffer 'D' and the purified DNA was recovered in 50 µl elution buffer. DNA extraction from the sediment samples removed from the artefacts manually or through water washes (sediment pellets) was performed using the same method but different input volumes. In brief, lysates were prepared by transferring up to 130 mg sediment to a 2-ml low-bind Eppendorf tube, adding up to 2 ml lysis buffer (0.45 M EDTA, pH 8.0, 0.05% Tween-20 and 0.25 mg ml$^{-1}$ proteinase K) (1 ml for samples of less than 100 mg or 0.5 ml for samples of less than 25 mg) and incubating overnight at 37 °C under rotation. DNA purification was performed using 500 µl or 1,000 µl lysate.

Negative controls containing sodium phosphate buffer or lysis buffer without sample material were carried alongside the samples through all steps of subsequent sample preparation and sequencing. For a subset of samples, DNA was extracted from only the first phosphate buffer fraction produced at each of the four incubation temperatures. Supplementary Data 1 provides an overview of the DNA extracts generated in this study.

## Library preparation

Of the extract, 10 µl was then converted into single-stranded DNA libraries using the automated protocol described in Gansauge et al. (2020)[23]. The number of unique library molecules obtained and the efficiency of library preparation were determined using two quantitative PCR assays[45]. Libraries were then amplified and double-indexed by PCR[46]. One library was prepared from each DNA fraction, except for the second and third 90 °C phosphate fractions from the DCP1 from which five libraries were prepared, each to maximize the yield of sequence data. Negative controls containing no sample material were carried along each set of extraction and library preparation. Note that there was substantial variation in library preparation efficiency across samples (Fig. 2 and Supplementary Data 1), ranging from 21.8% on average for the phosphate DNA fractions obtained from Bacho Kiro Cave specimens to 60.8% for DCP1 and 62.0% for the Quinçay specimens.

## Enrichment of mitochondrial and nuclear DNA by hybridization capture

To determine the taxonomic composition of the DNA recovered from the artefacts, libraries prepared from nearly all extracts generated in this study (all samples and controls, with the exception of the second and third phosphate fractions from BKP3) were enriched for mammalian mtDNA by hybridization capture with a probe set ('AA75') encompassing the mtDNA genomes of 242 mammalian species[24]. In cases in which several libraries were prepared from the same DNA extracts (that is, the 90 °C phosphate fractions from DCP1), only the first library was enriched for mammalian mtDNA. In addition, all libraries were enriched specifically for hominin mtDNA using a probe set ('AA163') designed in 1-bp tiling based on the revised Cambridge reference sequence of the human mitochondrial genome[32]. Both types of mtDNA captures were performed using two consecutive rounds of automated capture as detailed in Slon et al. (2017)[6] or Zavala et al. (2022)[47]. An overview of the capture reactions performed is provided in Supplementary Data 1.

In addition to mtDNA captures, 11 libraries prepared from the first, second and third 90 °C phosphate fractions of DCP1 were enriched for human nuclear DNA by two consecutive rounds of in-solution capture[48]. This enrichment was performed using a subset of a previously designed capture probe panel[8], named AA204, targeting two groups of SNPs: (1) 59,232 SNPs, randomly selected from set 6 'hominin diagnostic sites'[8], which represent positions in the genome where primates differ from other mammals and are used to quantify faunal mis-alignments; and (2) 411,492 SNPs selected from the '1240k' panel[30,49] (set 2 in Vernot et al. (2021)[8]). This set of SNPs is informative for investigating modern human population histories. Both SNP groups are located in regions of large evolutionary sequence divergence between humans and other mammals[8].

## Sequencing and raw sequence processing

Libraries enriched for mtDNA were combined into pools and sequenced on multiple lanes of a MiSeq sequencer (Illumina Technologies) in paired-end configuration with two index reads (2× 76 + 2× 8 cycles). Libraries enriched for human nuclear DNA were sequenced either on a HiSeq 2500 using the same configuration or on a HiSeq4000 (both Illumina Technologies) in single-read configuration with two index reads (1× 76 + 2× 8 cycles). In addition, one library prepared from the second 90 °C phosphate fraction of the DCP1 was sequenced directly (shotgun sequencing, without hybridization capture) on one lane of a HiSeq 4000 sequencer (Illumina Technologies) in single-read configuration. Base calling was performed using Illumina's Bustard tool and sequences were assigned to the library that they originated from, requiring perfect matches to the expected index combinations. LeeHom (https://github.com/mpieva/leeHom/tree/v.1.1.5)[50] was used to trim adapters and, for paired-end data, to merge overlapping paired-end reads.

## Taxonomic assignment of mtDNA sequences

Sequences resulting from mammalian or human mtDNA capture were assigned to mammalian taxa on the biology family level using a previously published computational pipeline[6] based on BLAST and MEGAN (version 0.0.12)[51], with modifications in data filtering detailed in Vernot et al. (2021)[8] (Supplementary Data 1 and Supplementary Information 4). In brief, as described in the latter study, false identifications of taxa were minimized by requiring at least three unique sequences (covering at least 105 positions in the reference genome) to be assigned to a family, and by requiring these sequences to represent at least 1% of all taxonomically identified sequences. The presence of ancient DNA was determined individually for the sequences from each family (and each library) by computing the frequency of terminal C-to-T substitutions as a proxy for the deamination rate at the molecule ends. Substitution frequencies significantly higher than 10% on both molecule ends (based on 95% binomial CIs) were then taken as evidence for the presence of ancient DNA from the respective family.

## Human and cervid mtDNA analysis

Present-day human contamination in the fractions from DCP1 that yielded ancient human mtDNA was estimated using the software tool AuthentiCT (version 1.0.0)[52] and a near-complete consensus sequence called for the second 90 °C fraction from DCP1 using positions with at least tenfold coverage of the mtDNA genome. This consensus sequence was then used for haplogroup assignment with Haplogrep 2 (version2.4.0)[53] and to identify 'diagnostic' positions in the mtDNA genome, which were used to determine the support for the consensus

sequence in each of the DNA fractions recovered from DCP1 (Extended Data Table 3). Tree building and genetic dating were performed using BEAST2 (version 2.6.6)[54]. Further information on the mtDNA analyses are provided in Supplementary Information 5. For genetic dating of the deer DNA component, near-complete wapiti mtDNA genomes were reconstructed from the first 60 °C phosphate fraction obtained from DCP1, as well as eight ancient wapiti samples as described in detail in Supplementary Information 6.

### Human nuclear DNA analysis

Human DNA capture data were processed as previously described[8] and data from the libraries with the lowest estimates of modern human contamination (based on AuthentiCT) were merged for further analyses. Principal component analysis including sequence data from present-day and other ancient human individuals was performed using smartpca (from EIGENSOFT package version 8.0.0)[55]. Using the R package admixr (version 0.7.1)[28], $f3$-statistics were calculated to determine shared genetic drift between DCP1 and a selection of modern and ancient human populations and their relationships were further evaluated using $D$-statistics. Sex determination for the human DNA component recovered from DCP1 was performed by comparing the coverage of the X chromosome and the autosomes in shotgun data obtained from the second 90 °C phosphate fraction after filtering against faunal mis-mappings[8]. Further details are provided in Supplementary Information 7.

### Reporting summary

Further information on research design is available in the Nature Portfolio Reporting Summary linked to this article.

### Data availability

The data that support the findings of this study are included in the Article and its Supplementary Information.

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

**Acknowledgements** We thank V. Slon, C. Hopfe, M. Peresani and W. Roebroeks for their contributions in an early exploratory phase of the project; P. Kosintsev for providing samples; A. Hübner for help with data analysis; B. Schellbach and A. Weihmann for help with sequencing; S. Nagel, J. Richter, B. Nickel and A. Aximu-Petri for help in the laboratory; J. Visagie for data processing; L. Jauregui, A. Bossoms Mesa, A. Pelin Sümer, H. Temming, A. Lister, M. Meiri and Z. Rezek for comments on the manuscript and/or logistical support; the National Museum of Natural History in Sofia for cooperating; R. Spasov for help with excavation at Bacho Kiro Cave; F. Lévêque, A. Lévêque, C.-H. Bachelier and J. Bachelier for facilitating and supporting research at Quinçay and Les Cottés; S. Rigaud for help with morphological identification of Quinçay samples; the Max Planck Society for funding; and the French Ministry of Culture for allowing and funding research at Les Cottés and Quinçay. The archaeological studies at Denisova Cave were funded by the Russian Science Foundation (no. 22-28-00049). M.Soressi is funded by the NWO VICI award (Neandertal Legacy VI.C.191.07). The radiocarbon dating work received funding from the European Research Council under the European Union's Horizon 2020 Research and Innovation Programme grant agreement no. 715069 (FINDER) to K.D. Work by E.I.Z. was partially funded by the Miller Institute for Basic Research in Science, University of California Berkeley.

**Author contributions** E.Essel, M.Soressi and M.M. designed the study. E.Essel, E.S.-K., H.F., M.Soressi and M.M. developed and tested methods. M.B.K., M.V.S., A.P.D., I.B., M.-C.S., T.T., N.S., E.Endarova, S.P.M., J.-J.H. and M.Soressi excavated, identified or characterized archaeological material and/or provided archaeological context and interpretation. E.Essel, E.S.-K., H.F., K.D. and A.S. performed laboratory experiments or collected data. E.Essel, E.I.Z., E.S.-K., H.F., B.V., K.D., M.Szymanski, J.K., S.P., M.H., M.Soressi and M.M. performed, aided in or supervised data analysis. E.Essel, E.I.Z., M.Soressi and M.M. wrote the manuscript with input from all other authors. E.Essel, E.I.Z., E.S.-K., M.M. and M.Soressi contributed equally to this work.

**Funding** Open access funding provided by Max Planck Society.

**Competing interests** The authors declare no competing interests.

**Additional information**
**Correspondence and requests for materials** should be addressed to Elena Essel, Marie Soressi or Matthias Meyer.

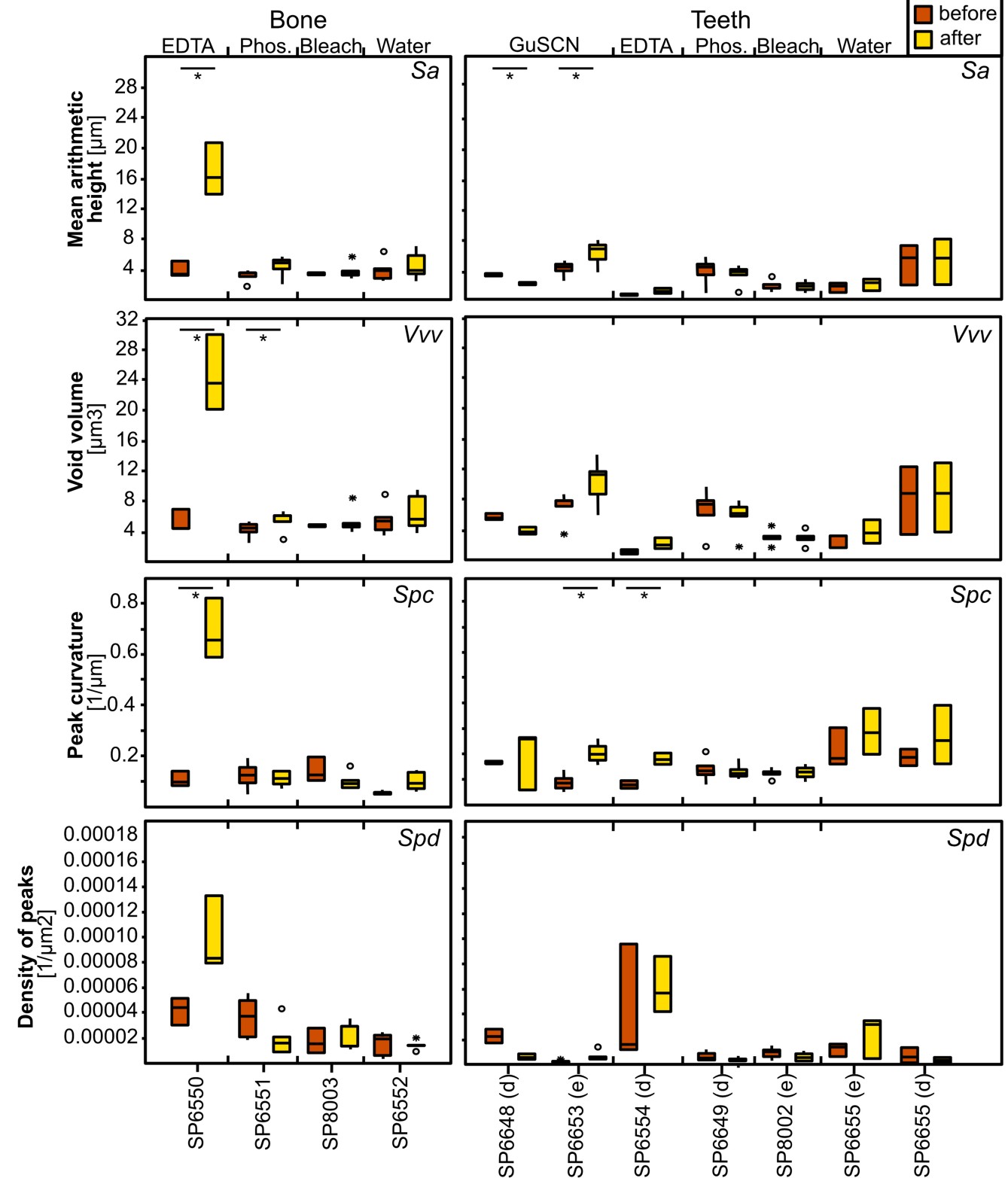

**Extended Data Fig. 1 | Microtopographic surface measurements.** Boxplots of the four selected surface texture parameters *Sa* (arithmetical mean height), *Vvv* (void volume of the valleys), *Spc* (arithmetic mean peak curvature), and *Spd* (density of peaks) by treatment (EDTA = ethylenediaminetetraacetic acid, GuSCN = guanidine thiocyanate reagent, Phos. = sodium phosphate buffer supplemented with detergent, Bleach = sodium hypochlorite, water = distilled water; e = enamel, d = dentine).

| Before DNA extraction | After DNA extraction | Before DNA extraction | After DNA extraction |
|---|---|---|---|
| SP6648 | GuSCN | SP6653 | GuSCN |
| SP6649 | Phosphate | SP6654 | EDTA |
| SP6650 | EDTA | SP6655 | Water |
| SP6651 | Phosphate | SP8002 | Bleach |
| SP6652 | Water | SP8003 | Bleach |

**Extended Data Fig. 2 | Faunal remains used in reagent testing.** Photographs of samples taken before and after treatment with various reagents used in ancient DNA extraction (GuSCN = guanidine thiocyanate reagent, EDTA = ethylenediaminetetraacetate solution, Phosphate = sodium phosphate buffer with detergent, Bleach = sodium hypochlorite solution). The black bar represents 1 cm. Note that colors are not directly comparable, as the photographs were taken at slightly different angles, with different light settings and camera adjustments.

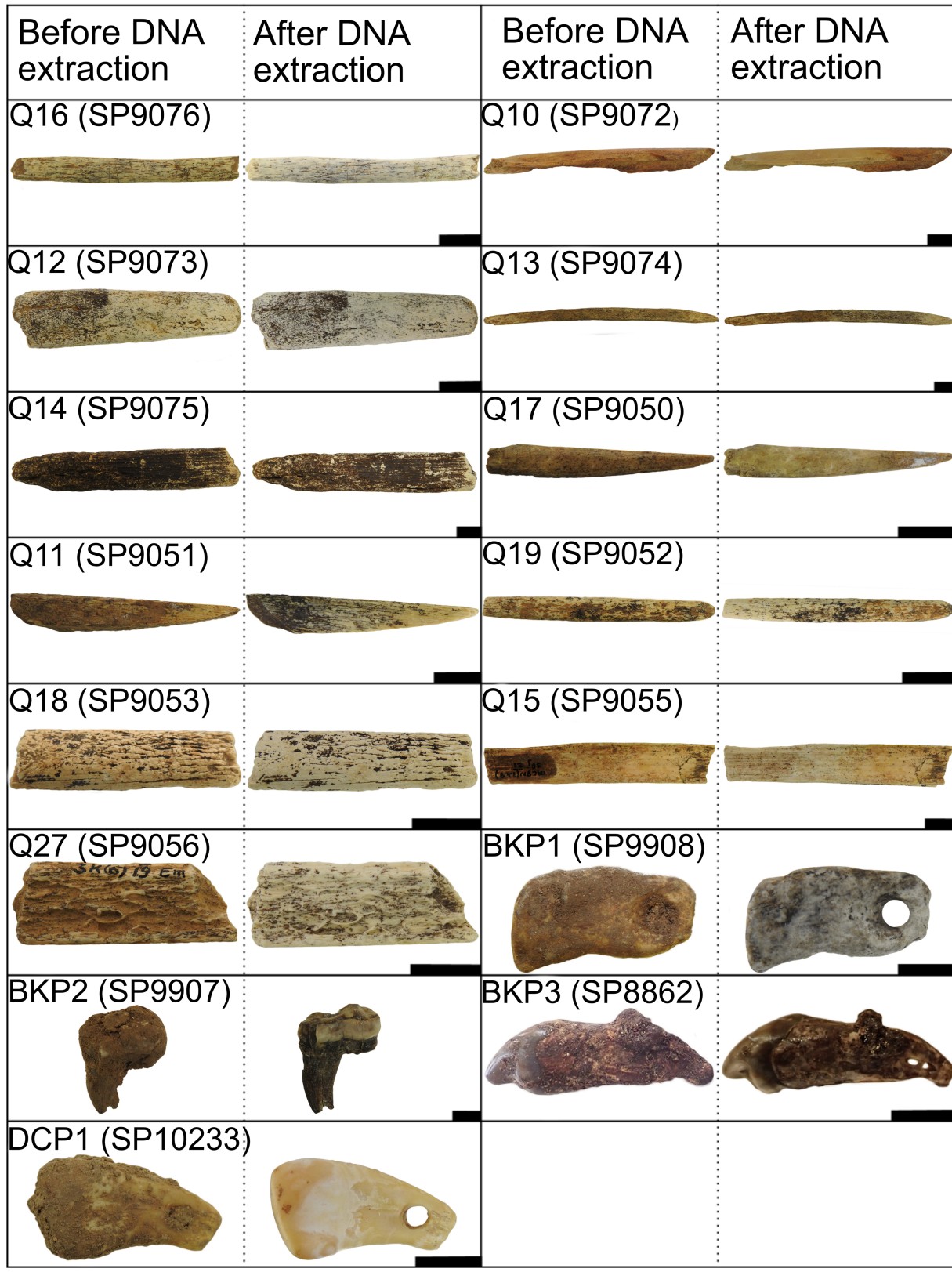

**Extended Data Fig. 3 | Artefacts before and after DNA extraction.**
Photographs of samples taken before and after phosphate-based, non-destructive DNA extraction. The black bar represents 1 cm. Note that colors are not directly comparable, as the photographs were taken at slightly different angles, with different light settings and camera adjustments.

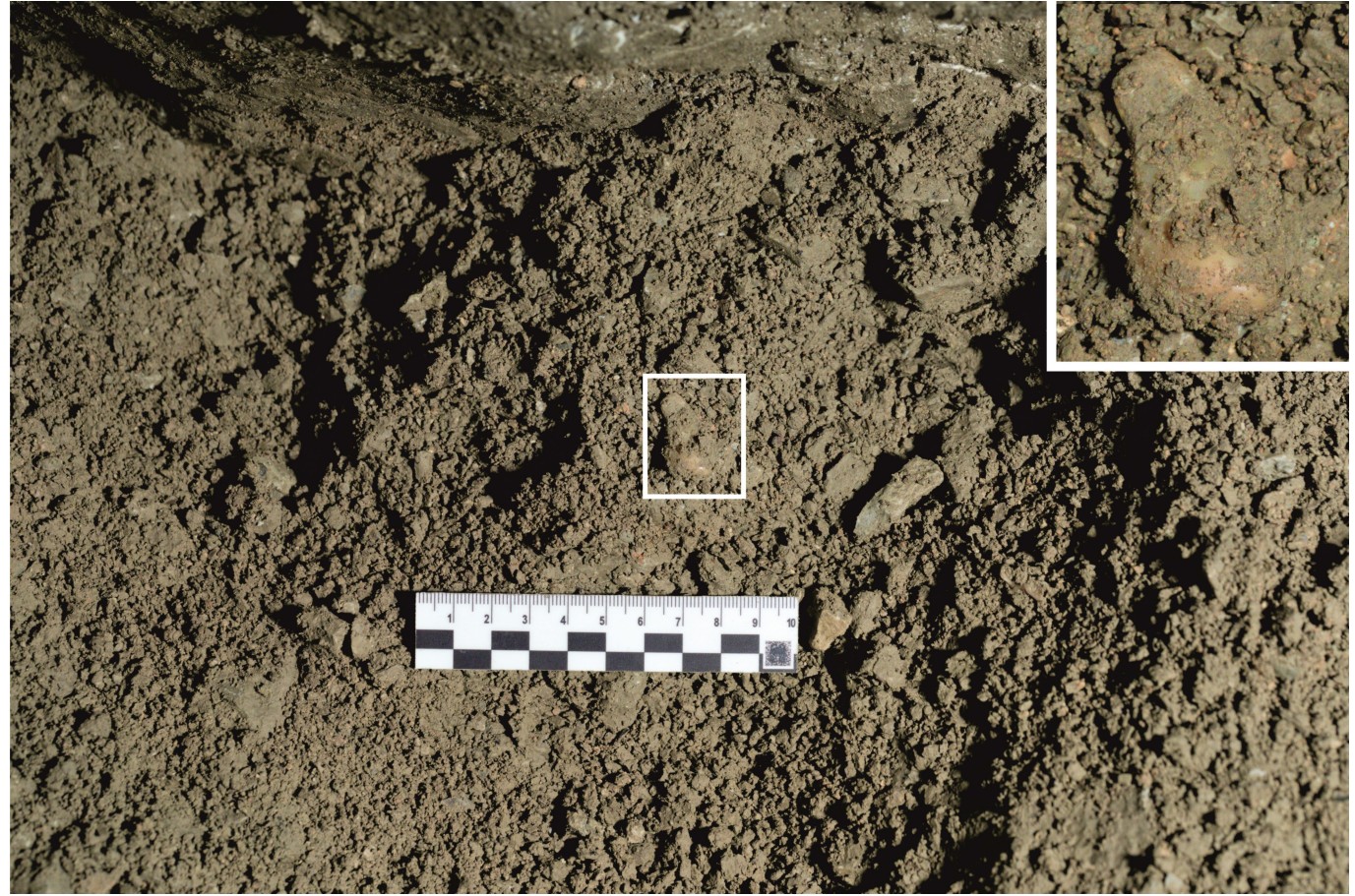

**Extended Data Fig. 4 | Photograph of DCP1 as it became exposed during excavation.** The photograph was taken shortly before the pendant was removed and placed into a plastic bag using gloves.

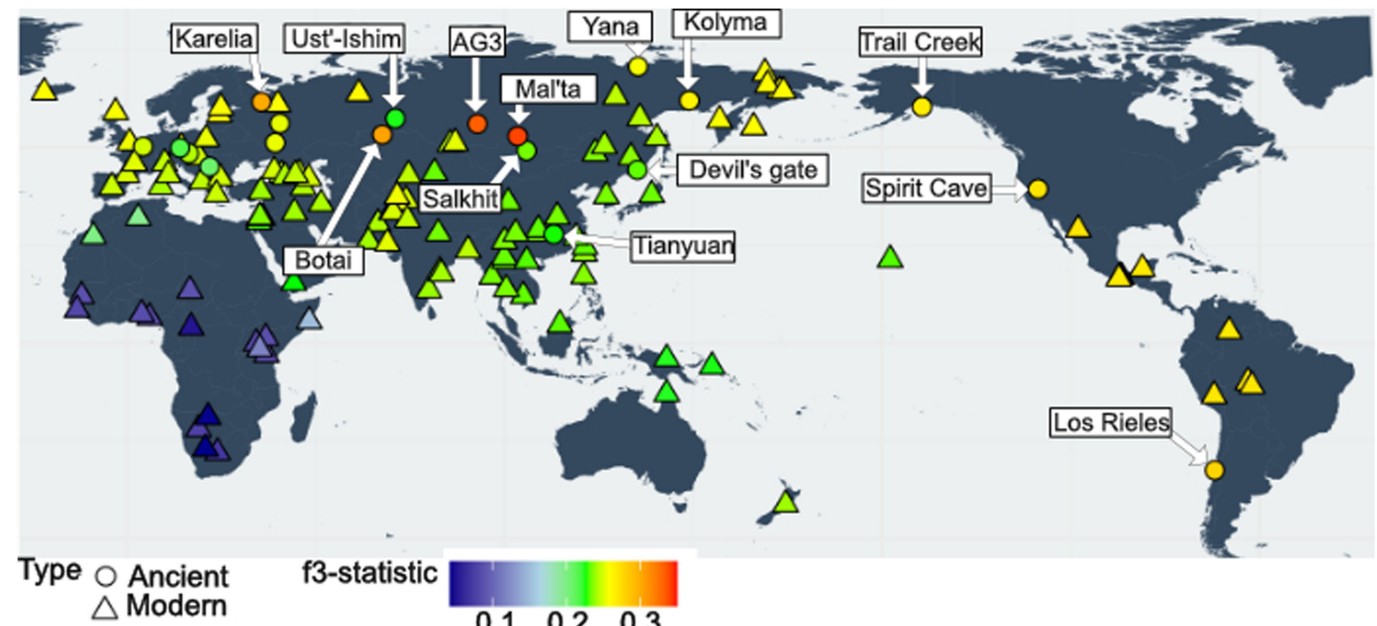

**Extended Data Fig. 5 | Shared genetic drift between DCP1 and different present-day and ancient human individuals.** Each point represents a distinct statistic using the calculation f3(X,Y; Mbuti) where Mbuti serves as an outgroup, population X is an ancient or present-day human population and Y is DCP1. Warmer colors on the map[56] represent more shared genetic drift.

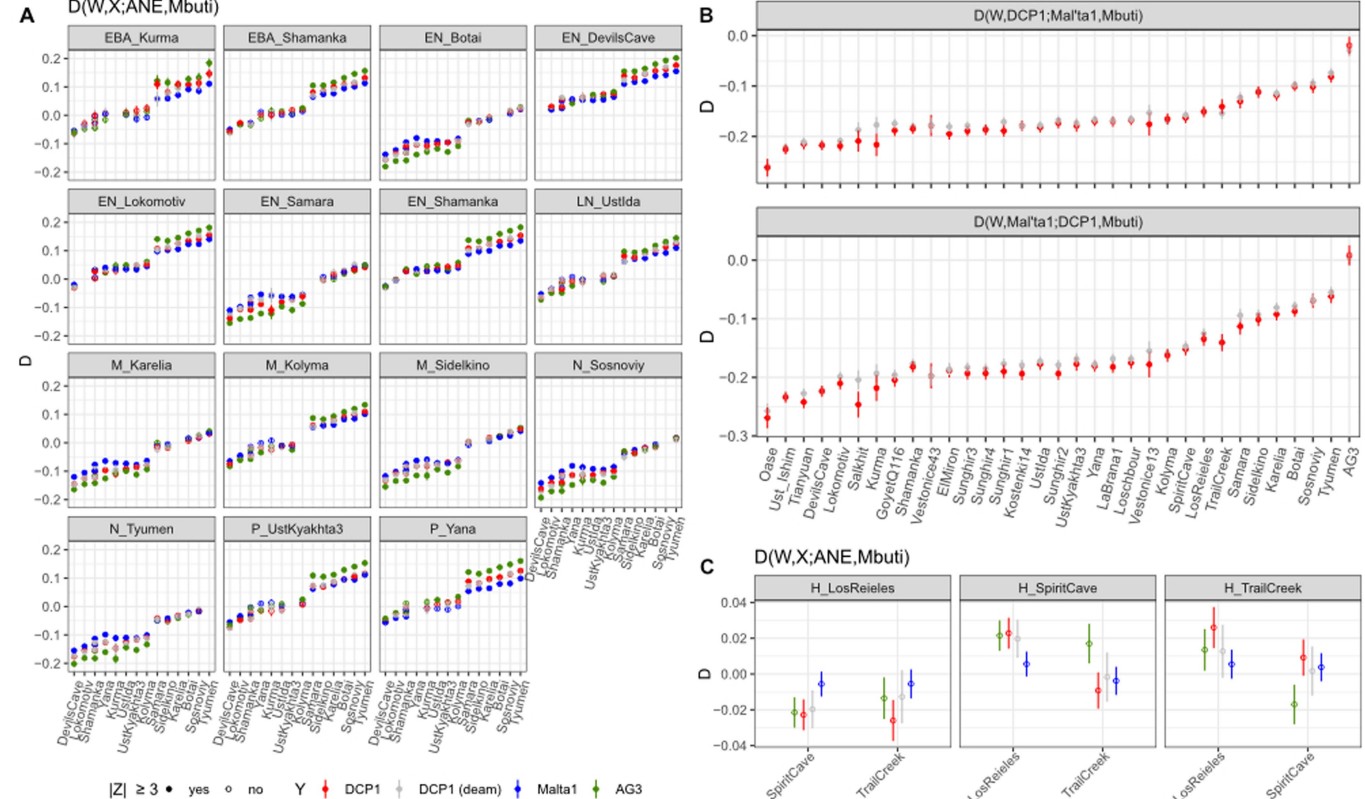

**Extended Data Fig. 6 | Genetic affinity of DCP1 to the genomes of other modern and ancient humans. (A)** Genetic affinities of DCP1 and Ancient North Eurasian (ANE) individuals (Y) to different ancient Siberians (W and X) based on 9,297-787,534 SNPs from the "1240k" SNP panel. **(B)** D-statistics comparing the genetic affinity of DCP1 and Mal'ta1 to a selection of ancient modern humans based on 13,618-237,630 SNPs from the "1240k" SNP panel. Red color indicates all sequences, grey color deaminated sequences only.

**(C)** Genetic affinities of DCP1 and ANE individuals (Y) to different ancient Americans (W and X) based on 42,548-574,391 SNPs from the "1240k" SNP panel. The calculations were performed using ADMIXTOOLS (version 5.1) via admixr (version 0.7.1) using all fragments or deaminated fragments only for DCP1. The error bars represent one standard error as calculated by a Weighted Block Jackknife. AG3 = Afontova Gora 3.

**Extended Data Table 1 | Description of the samples used in reagent testing and bone/tooth artefacts used for non-destructive DNA extraction**

| ID | Site | Type of sample | Morphological identification* | Estimated age | Excavation year | Context information | Reagent used ** |
|---|---|---|---|---|---|---|---|
| SP6648 | Les Cottés | Fragment of unmodified ivory | Mammoth/ Elephantidae | 55-35 ka | 2018 | Reworked sediment | GuSCN |
| SP6649 | Les Cottés | Carnivore canine unmodified | Carnivore | 55-35 ka | 2018 | Reworked sediment | Phosphate |
| SP6650 | Les Cottés | Rabbit or hare long bone unmodified | Rabbit/hare | 55-35 ka | 2018 | Reworked sediment | EDTA |
| SP6651 | Quinçay | Herbivore rib fragment unmodified | Herbivore | 45-35 ka | 1971-1980 | Layers Em/Ej | Phosphate |
| SP6652 | Quinçay | Mammal shaft unmodified | Mammal | 45-35 ka | 1971-1980 | Layers Em/Ej | Water |
| SP6653 | Quinçay | Herbivore tooth unmodified | Herbivore | 45-35 ka | 1971-1980 | Layers Em/Ej | GuSCN |
| SP6654 | Quinçay | Carnivore canine unmodified | Carnivore | 45-35 ka | 1971-1980 | Layers Em/Ej | EDTA |
| SP6655 | Quinçay | Carnivore canine unmodified | Carnivore | 45-35 ka | 1971-1980 | Layers Em/Ej | Water |
| SP8002 | Quinçay | Medium size herbivore incisor unmodified | Medium sized herbivore | 45-35 ka | 1971-1980 | Layers Em/Ej | Bleach |
| SP8003 | Quinçay | Medium size herbivore rib fragment unmodified | Medium sized herbivore | 45-35 ka | 1971-1980 | Layers Em/Ej | Bleach |
| Q17 (SP9050) | Quinçay | Modified bone | Medium to large mammal | 45-35 ka | unknown | Ej or Em | Phosphate |
| Q11 (SP9051) | Quinçay | Modified bone | Medium to large mammal | 45-35 ka | 1980 | Sfs | Phosphate |
| Q19 (SP9052) | Quinçay | Modified bone | Indeterminate | 45-35 ka | 1975 | Sm | Phosphate |
| Q18 (SP9053) | Quinçay | Weathered ivory fragment | Indeterminate | 45-35 ka | 1988 | Sfs | Phosphate |
| Q15 (SP9055) | Quinçay | Modified ivory | Indeterminate | 45-35 ka | 1990 | Sfs | Phosphate |
| Q27 (SP9056) | Quinçay | Unmodified bone fragment | Large mammal | 45-35 ka | 1970 | Em | Phosphate |
| Q10 (SP9072) | Quinçay | Unmodified bone fragment | Reindeer | 45-35 ka | 1972 | Emf | Phosphate |
| Q12 (SP9073) | Quinçay | Weathered bone or antler fragment | Indeterminate | 45-35 ka | 1972 | Emf | Phosphate |
| Q13 (SP9074) | Quinçay | Weathered bone fragment | Large ungulate | 45-35 ka | 1972 | Em eff. DIII | Phosphate |
| Q14 (SP9075) | Quinçay | Antler fragment | Indeterminate | 45-35 ka | 1978 | Sfj | Phosphate |
| Q16 (SP9076) | Quinçay | Modified bone or antler fragment | Indeterminate | 45-35 ka | 1971 | Ej-m | Phosphate |
| BKP1 (SP9908) | Bacho Kiro | Ornament | Large mammal | 45-43 ka | 2019 | Layer N1-I | Phosphate |
| BKP2 (SP9907) | Bacho Kiro | Ornament | Ursid | 45-43 ka | 2019 | Layer N1-H/I | Phosphate |
| BKP3 (SP8862) | Bacho Kiro | Ornament | Ursid | 45-43 ka | 2018 | Layer I/J | Phosphate |
| DCP1 (SP10233) | Denisova | Ornament | Cervid | 38-24 ka | 2019 | South Chamber, Layer 11/ level 2 | Phosphate |

Supplementary information 1 provides a detailed description of the samples.

* Morphological identifications were made through visual examination before or after cleaning the objects, independent of the results of DNA analysis.

** GuSCN = guanidine thiocyanate reagent, Phosphate = sodium phosphate buffer with detergent, EDTA = ethylenediaminetetraacetate solution, Bleach = sodium hypochlorite solution.

**Extended Data Table 2 | Differences between the mtDNA consensus sequence of DCP1 and other human mtDNA genomes**

| Position (in rCRS coordinate space) | 311 present-day humans | | 90 °C fraction #2 | | | Remark |
|---|---|---|---|---|---|---|
| | Dominant state | Prevalence (%) | Consensus base | Coverage | Consensus support [%] | |
| 3,107 | - | 100.0 | C | 56 | 91.1 | Insertion, excluded in further analyses |
| 4,826 | C | 100.0 | T | 35 | 74.3 | |
| 8,715 | T | 100.0 | C | 35 | 82.9 | |
| 9,156 | A | 99.7 | G | 52 | 94.2 | |
| 9,214 | A | 100.0 | G | 35 | 77.1 | |
| 9,681 | A | 100.0 | G | 42 | 88.1 | |
| 10,903 | C | 100.0 | T | 30 | 93.3 | |
| 12,237 | C | 100.0 | T | 54 | 90.7 | |

Listed are the positions determined to be 'diagnostic' for the consensus sequence obtained from the second 90 °C phosphate fraction of DCP1.

**Extended Data Table 3 | Support for the mtDNA consensus sequence reconstructed from DCP1**

| Fraction | ----------------------- All fragments --------------------- | | | ------------- Deaminated fragments ------------- | | |
|---|---|---|---|---|---|---|
| | Supporting 90 °C, fraction 2 | Supporting ≥99% of 311 present-day humans | % Supporting 90 °C, fraction 2 (95% C.I.) | Supporting 90 °C, fraction 2 | Supporting ≥99% of 311 present-day humans | % Supporting 90 °C, fraction 2 (95% C.I.) |
| Sediment pellet 1 | 4 | 16 | 20.0 (5.7-43.7) | 1 | 3 | 25.0 (0.6-80.6) |
| Sediment pellet 2 | 2 | 19 | 9.5 (1.2-30.4) | 0 | 1 | 0.0 (0.0-97.5) |
| 37 °C, fraction 1 | 0 | 0 | N/A (NA-NA) | 0 | 0 | N/A |
| 60 °C, fraction 1 | 3 | 5 | 37.5 (8.5-75.5) | 0 | 1 | 100.0 (0.0-97.5) |
| 90 °C, fraction 1 | 7 | 2 | 77.8 (40.0-97.2) | 0 | 1 | 100.0 (0.0-97.5) |
| **90 °C, fraction 2** | **243** | **37** | **86.8 (82.2-90.5)** | **54** | **9** | **85.7 (74.6-93.3)** |
| 90 °C, fraction 3 | 360 | 75 | 82.8 (78.9-86.2) | 83 | 4 | 95.4 (88.6-98.7) |

Number and percentage of mtDNA fragments in each DNA fraction recovered from the pendant matching the consensus sequence reconstructed from the second 90 °C phosphate fraction (bold).

# Reporting Summary

## Statistics

For all statistical analyses, confirm that the following items are present in the figure legend, table legend, main text, or Methods section.

| n/a | Confirmed | |
|---|---|---|
| ☐ | ☒ | The exact sample size ($n$) for each experimental group/condition, given as a discrete number and unit of measurement |
| ☐ | ☒ | A statement on whether measurements were taken from distinct samples or whether the same sample was measured repeatedly |
| ☐ | ☒ | The statistical test(s) used AND whether they are one- or two-sided<br>*Only common tests should be described solely by name; describe more complex techniques in the Methods section.* |
| ☒ | ☐ | A description of all covariates tested |
| ☒ | ☐ | A description of any assumptions or corrections, such as tests of normality and adjustment for multiple comparisons |
| ☐ | ☒ | A full description of the statistical parameters including central tendency (e.g. means) or other basic estimates (e.g. regression coefficient) AND variation (e.g. standard deviation) or associated estimates of uncertainty (e.g. confidence intervals) |
| ☐ | ☒ | For null hypothesis testing, the test statistic (e.g. $F$, $t$, $r$) with confidence intervals, effect sizes, degrees of freedom and $P$ value noted<br>*Give P values as exact values whenever suitable.* |
| ☐ | ☒ | For Bayesian analysis, information on the choice of priors and Markov chain Monte Carlo settings |
| ☒ | ☐ | For hierarchical and complex designs, identification of the appropriate level for tests and full reporting of outcomes |
| ☒ | ☐ | Estimates of effect sizes (e.g. Cohen's $d$, Pearson's $r$), indicating how they were calculated |

*Our web collection on statistics for biologists contains articles on many of the points above.*

## Software and code

Policy information about availability of computer code

| Data collection | No software was used for the collection of data. |
|---|---|
| Data analysis | All software packages used for analysis are publicly available and cited in the Online Methods section or in the Supplementary Information. These include together with their applications:<br>BAM file handling: samtools (version 1.3.1),<br>Metagenomics analysis: BLAST and MEGAN (version 0.0.12, see Slon et al., 2017),<br>f3- and D-statistics: ADMIXTOOLS (version 5.1) and R package admixr (version 0.7.1),<br>Adapter trimming and overlap-merging of paired-end reads: leeHom (https://github.com/mpieva/leeHom/tree/v.1.1.5),<br>Contamination estimates: AuthentiCT (https://github.com/StephanePeyregne/AuthetiCT, version 1.0.0),<br>Haplogroup assignment: Haplogrep2 (version2.4.0),<br>Tree buiding and genetic dating: BEAST2 (version 2.6.6),<br>PCA: smartpca from EIGENSOFT package (version 8.0.0),<br>Radiocarbon date calibration: IntCal2020 and OxCal platform (version 4.4),<br>3DST analysis: Mountains Map® Premium, version 7.4.8076 (Digital Surf (Besançon, France)),<br>FTIR analysis: Resolution Pro software (Agilent Technologies, version 5.3.0.1964),<br>mtDNA sequence alignment: MAFFT (version 7.453),<br>Clock and tree model selection: BEAST2's MODEL_SELECTION package,<br>Combination of log and tree files: BEAST2's logcombiner2,<br>Tree annotation: BEAST2's treeannotator program,<br>Tree visualization: Figtree (v1.4.4, https://github.com/rambaut/figtree/),<br>Tree tip dating: BEAST2's Tracer program, |

Sequence mapping: Burrows-Wheeler Aligner (BWA, version 0.5.10-evan.9-1-g44db244),
PCR duplicate removal: bam-rmdup (https://github.com/mpieva/biohazard-tools/tree/v0.2-knowngood),
Nucleotide substitution model selection: jModelTest (version 2.1),
Identification of primate sequences: Kraken (version 1),
Genotype calling: bam-caller version 0.2 (https://github.com/bodkan/bam-caller, 369)

For manuscripts utilizing custom algorithms or software that are central to the research but not yet described in published literature, software must be made available to editors and reviewers. We strongly encourage code deposition in a community repository (e.g. GitHub). See the Nature Portfolio guidelines for submitting code & software for further information.

## Data

Policy information about availability of data

All manuscripts must include a data availability statement. This statement should provide the following information, where applicable:
- Accession codes, unique identifiers, or web links for publicly available datasets
- A description of any restrictions on data availability
- For clinical datasets or third party data, please ensure that the statement adheres to our policy

Supplementary Information is available for this paper.
Supplementary Data file 1. Overview of DNA lysates, extracts and libraries prepared in this study and results of the mammalian and human mtDNA captures.
Supplementary Data file 2. Overview and summary statistics of the human nuclear DNA captures targeting 470,724 positions in the genome.

Cervid mtDNA sequences used for probe design, mapping and phylogenetic reconstructions were retrieved from NCBI GenBank, under accession numbers AB245427, JN632610, NC_020700, NC_050863, MG020563-MG020567 and MG020569-MG020571.

The reconstructed ancient human and cervid mtDNA sequences can be found on Dryad at https://doi.org/10.5061/dryad.41ns1rnj1 along with the SNPs used in the nuclear DNA enrichment. Sequencing data have been deposited in the ENA under project number PRJEB56213.

## Human research participants

Policy information about studies involving human research participants and Sex and Gender in Research.

| | |
|---|---|
| Reporting on sex and gender | N/A |
| Population characteristics | N/A |
| Recruitment | N/A |
| Ethics oversight | N/A |

Note that full information on the approval of the study protocol must also be provided in the manuscript.

# Field-specific reporting

Please select the one below that is the best fit for your research. If you are not sure, read the appropriate sections before making your selection.

☒ Life sciences      ☐ Behavioural & social sciences      ☐ Ecological, evolutionary & environmental sciences

For a reference copy of the document with all sections, see nature.com/documents/nr-reporting-summary-flat.pdf

# Life sciences study design

All studies must disclose on these points even when the disclosure is negative.

| | |
|---|---|
| Sample size | Sample sizes were limited by the availability of suitable archaeological material for potentially destructive analyses. For testing DNA extraction reagents, two independent samples were used per reagent, and four measurements obtained per sample. The statistical power achieved in these experiments is reflected in the significance values. Only four cleanly excavated objects could be obtained for non-destructive DNA extraction. These objects were analyzed and characterized independently of each other; no data was combined for statistical testing. |
| Data exclusions | No samples were excluded from this study. |
| Replication | Technical experiments were replicated using two samples per condition. DNA library preparation was replicated for the two DNA extracts from DCP1 that yielded the highest number of ancient human DNA sequences in non-destructive DNA extraction (five libraries each extract, see Data file S1). |
| Randomization | Randomisation was not relevant to this study because the analysis of DNA sequence data performed here does not involve procedures that depend on subjective evaluation or are otherwise dependent on preconceived hypotheses. In addition, randomization and blinding are |

precluded by the unique shape of the objects analyzed.

| Blinding | All data come from archaeological objects that are unique in size and shape, making blinding impossible. |

# Reporting for specific materials, systems and methods

We require information from authors about some types of materials, experimental systems and methods used in many studies. Here, indicate whether each material, system or method listed is relevant to your study. If you are not sure if a list item applies to your research, read the appropriate section before selecting a response.

## Materials & experimental systems

| n/a | Involved in the study |
|-----|----------------------|
| ☒ | ☐ Antibodies |
| ☒ | ☐ Eukaryotic cell lines |
| ☐ | ☒ Palaeontology and archaeology |
| ☒ | ☐ Animals and other organisms |
| ☒ | ☐ Clinical data |
| ☒ | ☐ Dual use research of concern |

## Methods

| n/a | Involved in the study |
|-----|----------------------|
| ☒ | ☐ ChIP-seq |
| ☒ | ☐ Flow cytometry |
| ☒ | ☐ MRI-based neuroimaging |

## Palaeontology and Archaeology

| Specimen provenance | All permits for excavation at Denisova Cave were obtained by the Institute of Archaeology and Ethnography, Siberian Branch of the Russian Academy of Science from the Ministry of Culture of the Russian Federation, and molecular analyses of the material granted as part of an agreement of scientific cooperation between the Institute of Archaeology and Ethnography, Siberian Branch of the Russian Academy of Sciences and the Max Planck Institute for Evolutionary Anthropology for projects in the field of palaeogenetics in North Asia, signed on December 25, 2018 and valid for a duration of five years. Permits for excavations and molecular analyses at Bacho Kiro were provided by the Bulgarian Ministry of Culture and the National Museum of Natural History, Sofia, Bulgaria (permit for regular archaeological investigation №RAP-120-11, signed on May 5, 2019). Permits for excavation at Les Cottés and Quincay were provided by the French Ministry of Culture. Approval for molecular analysis was granted through agreements of cooperation between the Max Planck Institute for Evolutionary Anthropology and M. Soressi representing the owners of the Les Cottés and Quincay collections. |

| Specimen deposition | Specimens used in this study are currently stored at the Max Planck Institute for Evolutionary Anthropology and will be returned to the owning institutions after publication of the study. |

| Dating methods | Two new radiocarbon dates were obtained from charcoal samples from layer 11 in the South chamber of Denisova Cave. Pre-treatment was done with the AOx-SC method, pre-treatment and dating were performed at the Oxford Radiocarbon Accelerator Unit in Oxford, UK. All previously published radiocarbon dates were newly re-calibrated using IntCal20 (see Supplementary information). |

☒ Tick this box to confirm that the raw and calibrated dates are available in the paper or in Supplementary Information.

| Ethics oversight | Ethical and legal concerns associated with excavation, material transfer and molecular analyses were addressed as part of the permits and scientific cooperation agreements cited above (see 'Specimen provenance'). |

Note that full information on the approval of the study protocol must also be provided in the manuscript.

