## [Peer Review File · Nature]

Manuscript Title: Ancient human DNA recovered from a Palaeolithic pendant

Reviewer Comments & Author Rebuttals

Reviewer Reports on the Initial Version:

Referee #1 (Remarks to the Author):

This article describes the optimization of a non-destructive way to extract DNA from bone and teeth that was compared to a previously published method published by Hofreiter (2012). This study was divided into 3 phases: Phase I describes the impact of various reagents on the structure of 10 bones or teeth samples and sodium phosphate buffer was chosen for subsequent experiments. Phase II describes the testing of this buffer on 11 artefacts excavated years ago. Unfortunately, all the objects were highly contaminated with contemporary human and pig DNA. Phase III describes the application of the new protocol on 4 freshly excavated samples. This time, PPE was worn by people collecting samples in order to avoid contamination. One particular sample (DCP1), a tooth pendant discovered in the Denisovan cave, produced enough DNA to determine that the person wearing the ornament was a woman of mtDNA haplogroup U who shared strong genetic affinities to a group of Ancient North Eurasians. The pendant itself was identified as an elk/wapiti tooth.

This work is more important than ever as the number of human remains/artefacts available for genetic testing decreases and concerns regarding the ethics of destructing human samples increase. I really enjoyed reading this article and appreciated the fact that the main article was brief, and all the details were present in the Extended or Supplementary material.

My first question is about the protocol itself. I can't help wondering how much money all these experiments have cost and I believe very few laboratories could afford a similar project. Why didn't the authors perform a few more testing that would reduce the number of extracts? For example, could the three 30 minutes incubations per temperature be compared to a single 90-minute incubation thus reducing the number of fractions to 1 per temperature? I was very surprised that the release of the DNA was performed with no agitation. Surely a very gentle agitation would release more DNA without causing any visible damage. It's a pity that the impact of a gentle agitation was not tested in Phase I. Another option could be to pool all three fractions into one library to maximize DNA yield and lower cost and time. Or is the decision to have so many extracts simply an insurance that were some extracts lost, you would have spare material to work on?

I suggest the authors make the data available in ENA or GenBank. I must confess that before seeing the actual reads, I was skeptical about the results. This all seemed too good to be true. What eventually convinced me was 1) seeing the amount of deamination as well as the mtGenome profile and 2) the similarities between the age estimates for the human sequence, the wapiti sequence and the carbon dating of the coal. This is, indeed, very neat.

Talking about mtDNA, I was wondering why positions 4826, 9214 and 12308 were omitted in the human mtDNA consensus? From what I saw in the mapping, there doesn't seem to be any doubt about the authenticity of these SNPs and 12308, in particular, would strengthen the U haplogroup determination (see jpeg 12308). Was the consensus profile including 4826T, 9214G and 12308G

tested and how did that modify the phylogenetic tree? Since I mention the tree, I recommend the authors improve Figure 3A, as it is difficult to read anything.

Question about the program AuthenticT which I am not familiar with. Did you use this program to separate reads that show deamination from reads that don't? Or did you use a different program? I saw several reads in the deaminated fraction that did not show any sign of damage (see jpegs).

Maybe taking those undamaged reads away would have changed your analyses.

While this protocol will certainly be useful to obtain DNA without damaging samples, couldn't it also be used as a pre-test to select samples that may contain endogenous DNA? I was just wondering if you had tried to powder the samples in order to perform a traditional DNA extraction after the non-destructive extractions. How would the DNA yield in a traditional extract compare to the yield obtained with the 90°C fractions? Wouldn't it be useful to compare what you got with the nondestructive method to what a traditional (destructive) protocol would produce? For example, for BK1, only 1410 reads were used to suggest that the bone came from a bovid. Is that truly enough to determine the origin of the bone? How can you exclude a potential bovid contamination from the sediment that actually contained 39,321 bovid reads?

Talking about mammalian species determination, it would be nice if you could clarify if a species/family could be attributed to any of the sample you tested before your experiments started. For example, Q15 obviously comes from an Elephantidae. Could any other species be identified by the people who discovered them? Either through a visual examination or by using contextual information about the dig (other remains found nearby, etc..).

I have a few simple comments/suggestions that the authors can take into consideration or ignore.

-Line 42: which are rare. Shouldn't it be which is rare? Or which is a rare event?

-Line 55: palimpsest? The definition I found is: a manuscript or piece of writing material on which the original writing has been effaced to make room for later writing but of which traces remain.

Wouldn't "a multitude" be more accurate?

Line 62: because they are conducive to the penetration of body fluids due to their porosity => [because of their porosity which is conducive to the penetration of body fluids. Furthermore, they contain hydroxyapatite.....]

Line 75: I don't like the term "worked bones". Here are some suggestions to replace worked: modified; converted; reshaped; customized.

-Line 100: I would add "11 osseous objects, labeled Q1 to Q15, that were excavated..."

-Line 107: "mtDNA fragments in the 60 and 90 °C fractions of a bone object". Add Q10 for clarity.

-Line 111: same as previous comment, add Q15.

-Line 124: if the origin of these teeth could be determined, please add it there (if not the exact species, maybe at least the family). Did you know that DCP1 came from a cervid before getting DNA?

Line 104 in the supp material suggest that you did. I think it's important to show that your analyses confirmed the taxonomic assessments.

-Line 138: about the potential low library prep efficiency: couldn't the severity of the inhibition be assessed with the IPC after the qPCR?

-Line 142: add sample names, i.e ursid (BK2 and BK3), bovid (BK1) and cervid (DCP1).

-Line 157: The largest numbers of deaminated human mtDNA fragments were obtained from the first libraries of the second and third..]

-Line 211: You need to refer to Table S2 there. I'm curious what % of the reads were characterized as human reads in the shotgun library.

-Line 211: add a reference for the method used to determine sex.

Methods

I'm not crazy about the term "technical experiments". I would call it Phase I or Reagents testing.

-Line 403: out of curiosity, why do the incubation in GuSCN have to be performed in the dark?

-Line 481: do you mean that if the volume after centrifugation is under 300 μ l, you qsp with TE so that all final volumes are 300 μ l? Why TE? Wouldn't Tris HCl or at least TLE be better to avoid downstream inhibition?

-Line 486: do you use the Zymo Spin columns? It seems that few laboratories still use them.

Figures. Extended and supplementary material

I'm not sure all the geological information is necessary in this article.

-Figure 2: why is DCP1 displayed twice? Especially since they do not match each other. Shouldn't the first DCP1 be BKP1?

-Figure 3 A and C are both too hard to read. As are all the extended data figures 5.

-Extended data table 2. If the animal species used as an object was known, add it there.

-Line 447: position 514-515 should be 523-524.

-Line 456. I don't understand. 12308G is definitively there.

-Line 464: how do you isolate reads that cover a specific position? Do you have a program for that?

Line 573: maybe you should mention that a wapiti is an elk native to North America and East Asia for those not familiar with the word wapiti.

-Line 653: extended data table 3: the 100% (for 3107) is misaligned.

-Supp material, line 135 "Faunal remains". I think lines 136 to 145 should be moved to the main text. Table S2 needs a legend specifying that these are results for DCP1 and remind how many SNP were targeted. Colum J: duplication rate seems too low. How did you come up with these numbers?

Referee #2 (Remarks to the Author):

A. Summary of the key results

The authors present an innovative method for extracting DNA from archaeological objects made of bone or tooth. These remains are a previously untapped source of ancient human DNA that can provide insights about the ancestry and biological sex of the individuals who handled, carried or wore these objects in the deep past. The non-destructive DNA extraction method reported here allows a stepwise release of this DNA, making it possible to distinguish DNA that penetrated deeply into an object during its manufacture or use from DNA that may originate from the surrounding sediment.

B. Originality and significance

This new approach opens up extremely interesting perspectives, particularly concerning sites where it is important to determine the identity of the site's occupants. I am thinking particularly of sites of the transition between the Middle and Upper Paleolithic for which we do not know whether Homo sapiens or Homo neanderthalensis occupied them.

Furthermore, we know that there was a division of labor in the settlements but we do not know how these tasks were divided between men and women. The analysis of ancient DNA contained in bone or ivory tools will allow a better understanding of the social organization of these ancient

populations.

C. Data & methodology

Being an archaeologist, I cannot comment on the validity of the approaches and the quality of the data, for that you need to have the text read by a geneticist.

D. Appropriate use of statistics and treatment of uncertainties

Same remark.

E. Conclusions: robustness, validity, reliability

As a prehistorian, I am totally convinced of the seriousness of this approach and of its importance for the scientific community. However, I wonder about the feasibility of the method on a large scale and its cost. It would be good if the authors could say something about this in their conclusion.

F. Suggested improvements: experiments, data for possible revision

Same answer as for questions C and D.

G. References

About DNA analyses in sediments, the authors could cite the recently published paper by Kurt H. Kjær et al. (Nature, vol. 612, pages 283-291, 2022) on 2-million-year-old sediments in Greenland. This is consistent with the idea that these analyses are just beginning and will improve over time to provide us with more and more information about past environments and human and animal populations.

H. Clarity and context

The abstract is clear and appropriate, as are the introduction and conclusions. The authors could just emphasize a bit the interest of such studies in places and periods where two populations have cohabited in order to better understand their interactions.

Prof. Sophie A. de Beaune

Referee #3 (Remarks to the Author):

A. The key results of the manuscript consist of the non-destructive extraction of ancient human DNA from a deer pendant found in layer 11 of the Denisova Cave, with a genetic age estimate of 20.2 (range 6.1-33.4 ka) and an attribution to a female individual, genetically close to other Paleolithic human remains, dated to ca. 24 ka and 17 ka, also found in Siberia but further East. Other key findings are the comparisons between the age estimates obtained via the ancient human genetic age attribution of the pendant, the ancient deer genetic age attribution of the pendant and the radiocarbon dates on other objects found close by in the same archaeological layer.

B. The contribution is original, as it presents for the first-time ancient DNA recovered from a Paleolithic artefact, a perforated deer tooth, i.e. an item known to be used as personal

body/garment decoration. Identifying biological sex and genetic ancestry of such an object is significant for the discussions on the origin(s) of personal adornment, symbolism, the trajectories of cultural traditions and on the identification of sex specific material culture in prehistoric societies.

An important element of the contribution is also the elaboration of a non-destructive method for DNA extraction, which as such is not new (cf. references provided by the authors) but very welcome for archaeological research questions addressing rare and unique samples. Furthermore, the comparison of genetic age estimates from human and faunal aDNA recovered from the same object is highly interesting, even if the range of these estimates is huge and dependent on a number of assumptions regarding population sizes and reproduction rates. The comparison with age estimates obtained via radiocarbon dating is only indirect as it does not apply to the same object, but to objects found close by (a direct radiocarbon date on the deer pendant itself would have been most interesting as well). Those indirect radiocarbon ages are quite different in age from each other. The implications of these discrepancies are not fully discussed.

C/F. The justification and followed procedure for establishing the non-destructive DNA isolation method are explicit but not 100% convincing and could benefit from some more details/explanations.

Of the 4 tested reagents, two were excluded because of their possible impact on the surface topography of the sample and one was retained because it offers the possibility of temperature-controlled DNA release. The usefulness of temperature-controlled DNA release is however not explained and also the reason to exclude the fourth reagent is not given.

Furthermore, the testing of the degree of surface modifications induced by the 4 reagents is not straight forward as the different reagents were applied to different types of raw material (e.g. mammoth ivory + herbivore tooth for GuSCN while carnivore canine + rabbit/hare long bone for EDTA), in addition from different stratigraphic layers and archaeological sites, and for which the comparative states of preservation (e.g. altered or well preserved) and types of surfaces (e.g. smooth or porous, encrusted with adhering exogenous material or not) are not given and for which the initial surface conditions may be different. A testing of the impact of different reagents on identical modern samples and on bones or tooth of the same species from a single chronologically and spatially homogeneous Paleolithic layer and previously cleaned (e.g. in an ultrasonic bath) would have been more convincing.

The non-destructiveness of the method is important if the research question requires an extensive number of samples or if destructive methods would cause a non-acceptable loss of information. If the research question is to identify the best way to extract DNA of the maker/user of a Paleolithic pendant, a destructive micro-sampling of a Paleolithic pendant, repeatedly done in the past for radiocarbon dating, seems justifiable especially given the fact that it is only partially destructive and that methods exist to document the outer topography and inner structure of such objects prior to destruction.

As many factors contribute to DNA preservation and extraction (e.g. Latham KE, Miller JJ. *Forensic Sci Res.* 2018) the non-destructiveness of the method seems subordinate and feasibility/reliability

tests of different methods should primarily be concerned with accuracy and alternative non-destructive methods could subsequently be compared.

The relevance of the application of the chosen non-destructive DNA isolation method to a series of different sorts of bones (modified and non-modified, well and badly preserved, different raw materials) from ancient excavations curated for a long time is not clear. These show evidence of contaminations, a fact well known already.

The conclusion that extended incubation at high temperature best enables the extraction of ancient human DNA could be reinforced if tests were performed for each temperature on the same number of incubations (if understood correctly, the 90°C incubations already passed three incubations for each of the lower temperature settings so a total of 9 previous incubations while the lower temperature settings passed much less incubations).

G/H. The abstract, introduction and conclusions seem appropriate.

The main text and figures however often miss reference to the information and data given in the Supplementary materials which should be mentioned more appropriately.

-Title reflects only part, albeit the most salient aspect, of the research presented in the paper

-Affiliations of the authors: Numbers 6 and 12 refer to the same institution`

-Line 63: the e.g. mentioned in round brackets should come after "body fluids" (not after "porosity")

-Reference 12 is incomplete.

-Extended data table 1: The description of the samples seems minimalistic. Reference to Supplementary Information SI1 (giving some information of the stratigraphic provenance/dating of the samples) should be made here and also in the main text (Line 80; this would also resolve the problem that SI2 is cited before SI1 in the main text); also, the abbreviations used in the "Reagent tested" column should be explained in the legend.

-Extended data figure 1: the figures cuts (bottom right) the explanation of the colors (red and yellow) used.

-Supplementary Information: it is a bit confusing that the lines are numbered per page and not as one sequence throughout the whole SI document or at least as a sequence/SI.

-Supplementary Information SI1: information on the state of preservation of the samples should be systematically given for all samples, as this variable may be important for the interpretation of the results. Also, the Headings on page 4 ("Stratigraphic context", "Faunal remains") and page 5 ("Radiocarbon chronology") could mention the archaeological site at stake as it is not immediately clear to which of the multiple samples described in SI1 they refer. In the section "A freshly excavated pendant from Denisova Cave": does "in the roof of layer 11" (Line 4) mean "in the upper part of

layer 11" (idem Line 13: meaning of "in the roof"?). In the "Radiocarbon chronology" section: it should be specified which date comes from the middle and which from the upper part of layer 11 (does the younger date come from the upper part and the much older date from the lower part of layer 11? And in what position within layer 11 was the pendant found?).

-Extended data table 2: taxonomic identification would be welcome here as it is directly relevant for the interpretation of the DNA results. Also reference to SI1 seems necessary here as it is not clear to what the abbreviations in the "context information" column refer.

-Extended data table 3: where do the "311 present-day humans" data come from?

-Figure 2: the legend states the figure is concerned with 6 artefacts but the figure provides reference to 5 samples only (DCP1 is represented twice; the reason why is not explained or immediately clear).

-Extended data figure 4: the photo does not really give information on the location of the pendant in the Denisova Cave, rather it gives a sight of the sediment surrounding it. The legend should be adapted or spatial and stratigraphic context information should be given in the figure.

-Figure 3: The sources of the human DNA data are not mentioned here. Reference should be made to SI5 and SI7.

-Supplementary table 3.1: the reason why there is no date provided for most of the samples in this table labelled "Radiocarbon dated samples" is not clear.

12,240

12,260

12,280

12,300

12,320

12,340

12,360

12,380

NC_012920.1 ATGCCCCATGTCTAACAACATGGCTTTCTCAACTTTTAAAGGATAACAGCTATCCATTGGTCTTAGGCCCAAAGAAATTTTGGTGCAACTCCAAATAAAAGTAATAACCATGCACACTACTATAACCACCCTAACCCTGACTTCCC

Consensus ATG T C C C C A T G T C T A A C A A C A T G G C T T T C T C A A C T T T T A A A G G A T A A C A G C T A T C C A T T G G T C T T A G G C C C A A G A A T T T T G G T G C A A C T C C A A A T A A A A G T A A T A A C C A T G C A C A C T A C T A T A A C C A C C C T A A C C C T A A C T T C C C T A A T

Multiple sequence alignment showing individual reads aligned to the consensus sequence. The reads are color-coded by nucleotide: A (green), C (blue), G (red), T (black). The alignment shows high similarity between reads, with a notable red vertical spike at position 12,310 corresponding to the coverage bar chart. The reads are truncated at the ends, indicated by dashes.

2,200

2,220

2,240

2,260

2,280

rCRS_circular ATTAAGAAAGCGTTCAAGCTCAACACCCACTACCT - AAAAAATCCCAAACATATAACTGAACTCCTCACACCCAATTGGACCAATCTATCACCCCTATAGAAGA

Consensus ATTAAGAAAGCGTTCAAGCTCAACACCCACTACCT - AAAAAATCCCAAACATATAACTGAACTCCTCACACCCAATTGGACCAATCTATCACCCCTATAGAAGA

AAAAAAGCGTTCAAGCTCAACACCCACTACCT - AAAAAATCCCAAACATATAACT C ATACCTAATTGGACCAATCTATCACCCCTATAGAAGA
 ATTAAGAAAGCGTTCAAGCTCAACACCCACTACT TCTCAAACATATAACTGAACTCCTCACACCCAATTGGACCAATCT
 ATTAAGAAAGCGTTCAAGCTCAACACCCACTACCT - AAAAAATCTT TCACACCCAATTGGACCAATCTATCACCCCTATAAAA
 ATTAAGAAAGCGTTCAAGTTC TTACTT - AAAAAATCCCAAACATATAACTGAACTCCTCACACCCAATTGGAC
 AAAGTGTTTAAGCTCAACACCCACTACCT - AAAAAATCCCAAACATATAACTGAACTCT CCAATTGGACCAATCTATCACCCCTATAGAAGA
 AGTGTTC AAGCTCAACACCCACTACCT - AAAAAATCCCAAACATATAAC TCACACCCAATTGGACCAATCTATCACCCCTATAGAAGA
 ATTAAGAAAGCGTTCAAGCTCAACACCCACTACCT AAAAAATCC ATATAACTGAACTCCTCACACCCAATTGACCAATCTATCACCCCTATAGAAGA
 ATTAAGAAAGCGTTCAAGCTCAACACCCACTACCT - AAAAAATTTAAAT TTGAACTCCTCACACCCAATTGGACCAATCTATCACCCCTATA
 GAAAGCGTTCAAGCTCAACACCCACTACCT - AAAAAATTCCAAACATATAACTGAACTCCTCACATC
 TGTTC AAGCTCAACACCCACTACCT - AAAAAATTCCAAACA TACACCCAATTGGACCAATCTATCACCCCTATAGAAGA
 GAAAGCGTTCAAGCTCAACACCCACTACCT - AAAAAATCCCAAATATATAATT
 AGCGTTCAAGCTCAACACCCACTACCT - AAAAAATCCCAAATAT
 TTT - AAAAAATCCCAAACATATAACTGAACTCTTCACACCCAATTGGACCAATCTATCAC
 TCAACACCCACTACCT - AAAAAATCCCAAACATATAACTGAACTCCTGACACCCAATTAGA AAGA
 TCCCAAACATATAACTGAACTCCTCACACCCAATTGGACCAATTTA
 TTAACATATAACTGAACTCCTCACACCCAATTGGACCAATCTATCATCTTATA
 ATTAAGAAAGCGTTCAAGCTCAACACCCACTACCT - AAAAAAT A TAACTGAACTCCTCACACCCAATTGGACCAATCTATT
 GAAAGCGTTCAAGCTCAACACCCACTACCT - AAAAAATCTC TAACTGAACTCCTCACACCCAATTGGACCAATCTATT AA
 GCGTTCAAGCTCAACACCCACTACCT - AAAAAATTTT AACTGAACTCCTCACACCCAATTGGACCAATCTATT
 AGCGTTCAAGCTCAACACCCACTACCT - AAAAAATCCTA ATAACTGAACTCCTCACACCCAATTGGACCAATCTATTATT
 TCAAGCTCAACACCCACTACCT - AAAAAATCCCAAAT AGACCAATCTATCACCCCTATAGAAGA
 TGTTC AAGCTCAACACCCACTACCT - AAAAAATCCCAAACAT
 TGTTC AAGCTCAACACCCACTACCT - AAAAAATCCCAAACATA
 TAAGCTCAACACCCACTACCT - AAAAAATCCCAAACATATAACTGAACT
 TCCACTACCT - AAAAAATCCCAAACATATAACTGAACTCCTCACAC
 TCACTACCT - AAAAAATCCCAAACATATAACTGAACTCCTCACAC
 ACATTCACTACCT - AAAAAATCCCAAACATATAACTGAACTCCTCACACTC
 TTT - AAAAAATCCCAAACATATAACTGAACTCCTCACACCC
 ATCT - AAAAAATCCCAAACATATAACTGAACTCCTCACACCCAATTG
 TCTAAACATATAACTGAACTCCTCACACCCAATTGGACCAATC
 AATCCCAAACATATAACTGAACTCCTCACACCCAATTGGACCAATCTATTATT
 AACTGAACTCCTCACACCCAATTGGACCAATCTATCATC
 TCAAACATATAACTGAACTCCTCACACCCAATTGGACCAATCTATCACCCCT
 TATAACTGAACTCCTCACACCCAATTGGACCAATCTATCACCCCTATAA
 ATCCCAAACATATAACTGAACTCCTCACACCCAATTGGACCAATCTATCACCCCTATAGAAA

880 900 920 940 960 980 1,000 1,020 1,040 1,060

rCRS_circular

Consensus

31

Coverage

Author Rebuttals to Initial Comments:

Response to the referees' comments are indicated in blue.

Referees' comments:

Referee #1 (Remarks to the Author):

This article describes the optimization of a non-destructive way to extract DNA from bone and teeth that was compared to a previously published method published by Hofreiter (2012). This study was divided into 3 phases: Phase I describes the impact of various reagents on the structure of 10 bones or teeth samples and sodium phosphate buffer was chosen for subsequent experiments. Phase II describes the testing of this buffer on 11 artefacts excavated years ago. Unfortunately, all the objects were highly contaminated with contemporary human and pig DNA. Phase III describes the application of the new protocol on 4 freshly excavated samples. This time, PPE was worn by people collecting samples in order to avoid contamination. One particular sample (DCP1), a tooth pendant discovered in the Denisovan cave, produced enough DNA to determine that the person wearing the ornament was a woman of mtDNA haplogroup U who shared strong genetic affinities to a group of Ancient North Eurasians. The pendant itself was identified as an elk/wapiti tooth.

This work is more important than ever as the number of human remains/artefacts available for genetic testing decreases and concerns regarding the ethics of destructing human samples increase. I really enjoyed reading this article and appreciated the fact that the main article was brief, and all the details were present in the Extended or Supplementary material.

My first question is about the protocol itself. I can't help wondering how much money all these experiments have cost and I believe very few laboratories could afford a similar project. Why didn't the authors perform a few more testing that would reduce the number of extracts? For example, could the three 30 minutes incubations per temperature be compared to a single 90-minute incubation thus reducing the number of fractions to 1 per temperature?

We thank the reviewer for the in-depth evaluation of the manuscript, the many constructive comments and especially also for taking time for evaluating the raw data.

Reducing the number of incubation steps is an excellent suggestion and something we should consider in follow-up studies. However, as the phosphate reagent itself is rather inexpensive, cost savings are largest if the number of DNA extractions and library preparations is minimized. As the reviewer suggests below, this could be achieved by pooling aliquots of the three phosphate fractions obtained at each temperature for DNA extraction and library preparation to determine whether any human DNA was released. For the Quincay and two of the Bacho Kiro artefacts we used a slightly different strategy to reduce costs in that we prepared libraries from only the first fraction obtained at each temperature. All fractions were analyzed for BKP3, which was the first freshly excavated object analyzed in this study, and DCP1, which yielded ancient human DNA. Further work is needed to determine which of those strategies is more efficient and whether the number of fractions can be reduced.

I was very surprised that the release of the DNA was performed with no agitation. Surely a very gentle agitation would release more DNA without causing any visible damage. It's a pity that the impact of a gentle agitation was not tested in Phase I.

We were considering using gentle agitation during the incubation steps, but decided against it to minimize the risk of damaging the objects. We were particularly concerned that a combination of heat and agitation might cause some objects to fall apart due to the presence of internal micro-cracks, which are not always visible from the outside. However, we agree that the effect of agitation should be tested in subsequent optimizations of the non-destructive extraction method.

Another option could be to pool all three fractions into one library to maximize DNA yield and lower cost and time. Or is the decision to have so many extracts simply an insurance that were some extracts lost, you would have spare material to work on?

This is a good idea and something we will implement in future work. We did not use the entire DNA fractions for DNA extraction and library preparation in our present work and would not do so when pooling fractions in the future, so repeated library preparation from individual fractions remains possible after an initial screening phase.

I suggest the authors make the data available in ENA or GenBank. I must confess that before seeing the actual reads, I was skeptical about the results. This all seemed too good to be true. What eventually convinced me was 1) seeing the amount of deamination as well as the mtGenome profile and 2) the similarities between the age estimates for the human sequence, the wapiti sequence and the carbon dating of the coal. This is, indeed, very neat.

We greatly appreciate the reviewer's efforts in re-analyzing the raw data, which we provided upon request. It is frustrating that unlike dryad, which we used for deposition of the consensus sequences and multiple sequence alignments, ENA does not provide reviewer access to data. We decided to hold the data release until the completion of the review process to ensure all metadata associated with the submission (e.g., the study abstract) is up-to-date upon final release, and in case a reviewer requests additional data production. We might consider an immediate data release in the future.

Talking about mtDNA, I was wondering why positions 4826, 9214 and 12308 were omitted in the human mtDNA consensus? From what I saw in the mapping, there doesn't seem to be any doubt about the authenticity of these SNPs and 12308, in particular, would strengthen the U haplogroup determination (see jpeg 12308). Was the consensus profile including 4826T, 9214G and 12308G tested and how did that modify the phylogenetic tree?

We thank the reviewer for pointing this out. The three positions mentioned are the only ones that were covered by at least 10 fragments and where the consensus support was below 80% (as shown in Suppl. Fig. 5.1 and mentioned in SI text, section 5). Here and in the past, for example when working with sediment DNA, we used a very conservative threshold of 80% for consensus calling to ensure the highest possible consensus accuracy. However, we now realize that this was not made clear in the manuscript and we agree that the completeness of the consensus, especially at positions that are variable within humans, is important to avoid biases in tree building and branch-shortening analyses.

Since the consensus support at all three positions is only slightly below 80% (74% or higher, see ED Table 2), we now included them in the consensus sequence and updated the multiple alignment file (incl. the version deposited on dryad), the tree and the haplogroup analysis accordingly. Since two of the positions are unique mutations on the DCP1 branch, we also updated all analyses using diagnostic positions (with only minimal changes in results). The point estimate for the age of the human mtDNA changes slightly from 20.2 to 18.5 ka, but remains consistent with the date obtained for the wapiti mtDNA.

Since I mention the tree, I recommend the authors improve Figure 3A, as it is difficult to read anything.

This is true. We have adjusted the font sizes in all panels to improve readability.

Question about the program AuthenticCT which I am not familiar with. Did you use this program to separate reads that show deamination from reads that don't? Or did you use a different program? I saw several reads in the deaminated fraction that did not show any sign of damage (see jpegs). Maybe taking those undamaged reads away would have changed your analyses.

AuthenticCT was used to estimate human contamination from the distribution of the deamination signal in the data, but not for isolating individual reads, which was done using a custom script. We are not familiar with the software the reviewer used for alignment visualization, but it appears that in Screenshot 2 provided by the reviewer, reads that are aligned in reverse complementary direction are marked in red. As we used a strand-specific library preparation method, C to T substitutions (in the direction read by the sequencer) appear as G to A substitutions in these reads. When inspecting each individual sequence alignment in Screenshot 2, we identify at least one C to T substitution within the first three or last three alignment positions of each green read (displayed in the direction read) and at least one G to A at the termini of each red read (displayed in reverse complement). Screenshot 3 is a zoomed-out alignment view that displays substitutions by small rectangles. We noticed that all reads in this alignment view carry small shadings on at least one end. We suspect that these shadings are introduced by the software to mask substitutions that are located in close proximity to the end of the reads, thereby masking the deamination signal. We suppose the Screenshot 1 was included to demonstrate the validity of the 12308G consensus call (now included), which was made using all reads without filtering for deamination, hence a mixture of deaminated and non-deaminated reads are present.

In summary, we are unable to identify problems in deamination filtering based on the information provided by the reviewer (we also visually checked some of the alignments again using 'samtools tview' and did not encounter unexpected patterns). We would also like to note that deamination filtering did not turn out to be necessary for the most critical analyses (mtDNA consensus building and nuclear DNA analyses).

While this protocol will certainly be useful to obtain DNA without damaging samples, couldn't it also be used as a pre-test to select samples that may contain endogenous DNA?

We are hesitant to suggest this, as it will not be practical in most cases. First, the method is mostly suitable for rather small skeletal elements (teeth, phalanges or small bone fragments), as there are practical constraints on the size of objects that can easily be submerged in relatively small volumes of extraction buffer. Second, surface contamination can be a severe obstacle for the method as we have shown and most objects in existing collections were not cleanly excavated and handled using gloves and other precautions. For these reasons, minimally destructive micro-sampling approaches (drilling 5-10 mg powder after surface removal) would seem more appropriate when targeting endogenous DNA in ancient bones and teeth. For artefacts the situation is different, as we are primarily interested in the DNA of the user/wearer, which is an exogenous DNA component.

I was just wondering if you had tried to powder the samples in order to perform a traditional DNA extraction after the non-destructive extractions. How would the DNA yield in a traditional extract compare to the yield obtained with the 90°C fractions? Wouldn't it be useful to compare what you got with the nondestructive method to what a traditional (destructive) protocol would produce? For example, for BK1, only 1410 reads were used to suggest that the bone came from a bovid. Is that truly enough to determine the origin of the bone? How can you exclude a potential bovid contamination from the sediment that actually contained 39,321 bovid reads?

This is a good suggestion and an experiment that should be performed in future studies. It would be interesting to destructively sample the freshly excavated artefacts, and in particular DCP1, to compare endogenous (cervid) and human DNA yields to the non-destructive method. However, it is not trivial to come up with a good experimental strategy. Decisions need to be made about where exactly to sample an artefact (and how often), as both endogenous and exogenous DNA content presumably vary on a microscale. In addition, destructive sampling is probably best performed before non-destructive DNA extraction, which alters the DNA content in the artefact as a whole. These questions justify a dedicated technical follow-up study that is beyond the scope of this proof-of-principle paper.

Regarding BKP1, the reviewer makes a good point here. For the other artefacts, the dominant mammalian families in the 37, 60 and 90 °C phosphate fractions consistently agree with the morphological identification of the objects. However, BKP1 is morphologically indeterminate and the only case where the dominant family in the surrounding sediment matches the dominant family in the phosphate fractions. In this case, species identification based on genetic evidence alone may not be fully conclusive. We have rephrased the respective sentence in the main text and added a caveat to the end of SI4.

Talking about mammalian species determination, it would be nice if you could clarify if a species/family could be attributed to any of the sample you tested before your experiments started. For example, Q15 obviously comes from an Elephantidae. Could any other species be identified by the people who discovered them? Either through a visual examination or by using contextual information about the dig (other remains found nearby, etc..).

Morphological identifications were made for most artefacts used in the study (independent of the results of DNA analysis). This information was indeed missing and has now been added to Extended data table 1. We also updated the respective sections in the main text to contextualise the genetic and morphological findings.

I have a few simple comments/suggestions that the authors can take into consideration or ignore.

Line 42: which are rare. Shouldn't it be which is rare? Or which is a rare event?

We fixed this by removing the unnecessary word 'contexts'.

Line 55: palimpsest? The definition I found is: a manuscript or piece of writing material on which the original writing has been effaced to make room for later writing but of which traces remain. Wouldn't "a multitude" be more accurate?

We agree. Changed as suggested.

Line 62: because they are conducive to the penetration of body fluids due to their porosity => [because of their porosity which is conducive to the penetration of body fluids. Furthermore, they contain hydroxyapatite.....]

We have changed the sentence as follows for better readability: "In theory, such analyses are most promising for artefacts made from animal bones or teeth, both because they are porous and thereby conducive to the penetration of body fluids (e.g. sweat, blood or saliva), but also because they contain hydroxyapatite, which is known to adsorb DNA and reduce its degradation by hydrolysis and nuclease activity [9,10]."

Line 75: I don't like the term "worked bones". Here are some suggestions to replace worked: modified; converted; reshaped; customized.

We simply removed the word 'worked', because it is not needed in this context.

Line 100: I would add "11 osseous objects, labeled Q1 to Q15, that were excavated..."

We added the object labels as suggested.

Line 107: "mtDNA fragments in the 60 and 90 °C fractions of a bone object". Add Q10 for clarity.

Changed as suggested.

Line 111: same as previous comment, add Q15.

Changed as suggested.

Line 124: if the origin of these teeth could be determined, please add it there (if not the exact species, maybe at least the family). Did you know that DCP1 came from a cervid before getting DNA? Line 104 in the supp material suggest that you did. I think it's important to show that your analyses confirmed the taxonomic assessments.

Morphological attributions were indeed made for many but not all of the artefacts, and independently of the DNA results. We have now added these attributions to ED table 1 and included a reference to this table in the sentence.

Line 138: about the potential low library prep efficiency: couldn't the severity of the inhibition be assessed with the IPC after the qPCR?

We do indeed use the internal positive control to quantify library preparation efficiency, which provides a measure of inhibition. Library preparation efficiency is shown for each DNA fraction in Fig. 2.

Line 142: add sample names, i.e ursid (BK2 and BK3), bovid (BK1) and cervid (DCP1).

We now specify the morphological attributions of the objects in the main text.

Line 157: The largest numbers of deaminated human mtDNA fragments were obtained from the first libraries of the second and third..]

In our view, this would make the statement unnecessarily complicated. Additional libraries prepared from the second and third 90°C fraction of DCP1 are essentially technical replicates, which produced very similar numbers of DNA fragments compared to the initially produced libraries (1st library 90°C fraction 2: 971 fragments, additional libraries: 811 - 1,003 fragments each; 1st library 90°C fraction 3: 1,096 fragments, additional libraries: 1,122 - 1,278 fragments each). In contrast, a maximum of 155 fragments were obtained in any of the preceding fractions.

The reviewer's comment helped us to spot an error in Fig. 2, where we plotted the number of human mtDNA fragments without deamination filtering although we intended to show the number of deaminated fragments as referenced in the text. This has been corrected now. The updated plot shows even more pronouncedly the difference in DNA recovery between the 90 °C phosphate fraction and the preceding DNA fractions.

Line 211: You need to refer to Table S2 there. I'm curious what % of the reads were characterized as human reads in the shotgun library. Line 211: add a reference for the method used to determine sex

We have now added a reference to the supplement for details on the sexing. Since a mixture of cervid and human DNA is present in the library, it is not easily possible to determine the percentage of the human component alone. In fact, for the sexing, no mapping to the whole human genome was attempted. Instead, mappings were performed to ~3 million sites in the human genome where humans and other mammals are sufficiently divergent to identify human reads. Out of the 272,775,081 reads that were generated, 20,526 unique reads were retained in this analysis (~0.008%), but the actual proportion of human DNA in the library is likely to be orders of magnitudes higher. These numbers are also provided in the supplement (section 7), which is now referred to in the main text.

Methods

I'm not crazy about the term "technical experiments". I would call it Phase I or Reagents testing.

Changed to 'reagent testing' as suggested.

Line 403: out of curiosity, why do the incubation in GuSCN have to be performed in the dark?

We simply followed the method of Rohland et al. 2004 here, who performed this incubation in the dark. We do not know the reason for this either, if there is any.

Line 481: do you mean that if the volume after centrifugation is under 300l, you qsp with TE so that all final volumes are 300l? Why TE? Wouldn't Tris HCl or at least TLE be better to avoid downstream inhibition?

This is correct. TE is our standard storage buffer for DNA and contains only little EDTA (1 mM). EDTA does not interfere with subsequent DNA purification, which comes before library preparation. Note that the lysis buffer used for DNA extraction from bones and sediment contains 450 mM EDTA.

Line 486: do you use the Zymo Spin columns? It seems that few laboratories still use them.

For DNA extraction we were using columns from the High Pure Viral Nucleic Acid Large Volume Kit (Roche catalogue number 05114403001). These are also the columns suggested in the Rohland et al. 2018 paper, which is the up-to-date reference for the DNA extraction we used. We removed other references from the Methods section and adjusted the text to make clear that this is the protocol that was used.

Figures. Extended and supplementary material

I'm not sure all the geological information is necessary in this article.

There are different views on this among the authors. Since the information is in the supplement and can be easily skimmed, we decided to leave it there.

Figure 2: why is DCP1 displayed twice? Especially since they do not match each other. Shouldn't the first DCP1 be BKP1?

This was a copy paste error. Thanks for spotting it!

Figure 3 A and C are both too hard to read. As are all the extended data figures 5.

We adjusted the font size in these figures, reformatted them and improved the resolution for better readability. In addition, we created space by focusing only on the relevant part of the human mtDNA tree in panel A and converting panel D to an Extended data figure.

Extended data table 1. If the animal species used as an object was known, add it there.

This information has been added. In addition, we merged Extended data tables 1 and 2 to create space for another Extended data figure (formerly Fig. 3D).

Line 447: position 514-515 should be 523-524.

The sequence of the reference genome at positions 514-523 is CACACACACA. One CA dinucleotide is deleted in all sequenced fragments. Samtools tview shifts the deletions mostly to the left (positions 514-515). But 522-523 would also be correct, as we do not know which dinucleotide was deleted.

Line 456. I don't understand. 12308G is definitively there.

12308G was initially not called, because it is supported by less than 80% of the reads (78.3%). Following the reviewer's comment above, this and two other positions have now been included in the consensus sequence and the text amended accordingly.

Line 464: how do you isolate reads that cover a specific position? Do you have a program for that?

The '-L' option of 'samtool view' offers this functionality. To produce summary statistics, a custom script was used.

Line 573: maybe you should mention that a wapiti is an elk native to North America and East Asia for those not familiar with the word wapiti.

We added this explanation as suggested.

Line 653: extended data table 3: the 100% (for 3107) is misaligned.

Fixed. Thanks for spotting this. The numbering of extended data items changed during the revisions. This is now extended data table 2.

Supp material, line 135 "Faunal remains". I think lines 136 to 145 should be moved to the main text.

We would prefer to keep the archaeological information together in one place. Moving everything to the main text is impossible due to space constraints.

Table S2 needs a legend specifying that these are results for DCP1 and remind how many SNP were targeted. Colum J: duplication rate seems too low. How did you come up with these numbers?

We added legends to the two supplementary data files following the reviewer's suggestion and added information on the number of SNPs targeted. There are different stages in the analysis at which duplication rates can be computed. We re-formatted and updated Table S2 to only include duplication rates before the kraken-based taxonomic assignment of reads, now retaining the formula that was used in the cells. Additionally, we made some further edits to Table S2 to improve its readability.

Referee #2 (Remarks to the Author):

A. Summary of the key results

The authors present an innovative method for extracting DNA from archaeological objects made of bone or tooth. These remains are a previously untapped source of ancient human DNA that can provide insights about the ancestry and biological sex of the individuals who handled, carried or wore these objects in the deep past. The non-destructive DNA extraction method reported here allows a stepwise release of this DNA, making it possible to distinguish DNA that penetrated deeply into an object during its manufacture or use from DNA that may originate from the surrounding sediment.

B. Originality and significance

This new approach opens up extremely interesting perspectives, particularly concerning sites where it is important to determine the identity of the site's occupants. I am thinking particularly of sites of the transition between the Middle and Upper Paleolithic for which we do not know whether Homo sapiens or Homo neanderthalensis occupied them.

Furthermore, we know that there was a division of labor in the settlements but we do not know how these tasks were divided between men and women. The analysis of ancient DNA contained in bone or ivory tools will allow a better understanding of the social organization of these ancient populations.

C. Data & methodology

Being an archaeologist, I cannot comment on the validity of the approaches and the quality of the data, for that you need to have the text read by a geneticist.

D. Appropriate use of statistics and treatment of uncertainties

Same remark.

E. Conclusions: robustness, validity, reliability

As a prehistorian, I am totally convinced of the seriousness of this approach and of its importance for the scientific community. However, I wonder about the feasibility of the method on a large scale and its cost. It would be good if the authors could say something about this in their conclusion.

We thank Prof. de Beaune for the positive feedback and suggestions.

In this proof-of-principle study we were striving to study DNA release from some of the objects (especially DCP1) at the highest resolution possible, in order to investigate at which point in the process the various types of DNA are released and in which quantities (DNA from the surrounding sediment, from the animal, the user/wearer(s) and human contamination). However, for most of the objects (for example the old material from the Quincay collection) only four DNA fractions needed to be analysed to determine that no ancient human DNA could be recovered. Costs for screening samples with this method are thus approximately four times higher than those incurred with the analysis of ancient DNA in human skeletal remains, which is work that is performed at the scale of many hundred samples these days. It thus seems economically feasible in principle, for example if funding can be secured for a medium-sized grant, to screen dozens if not hundreds of objects for DNA preservation and then concentrate efforts particularly on those that yielded ancient human DNA. In addition, simplifications to the workflow are conceivable that would further reduce costs.

The biggest hurdle we currently face is the scarcity of cleanly excavated osseous objects. As pointed out in the conclusions, we hope that this study will raise awareness for the importance of clean

excavation protocols, which will be the basis for more systematic analyses of DNA from the users of Palaeolithic artefacts.

F. Suggested improvements: experiments, data for possible revision

Same answer as for questions C and D.

G. References

About DNA analyses in sediments, the authors could cite the recently published paper by Kurt H. Kjær et al. (Nature, vol. 612, pages 283-291, 2022) on 2-million-year-old sediments in Greenland. This is consistent with the idea that these analyses are just beginning and will improve over time to provide us with more and more information about past environments and human and animal populations.

As the geneticists on this paper are very active in developing better methods for the analysis of highly degraded DNA from difficult samples, we cherish the idea that future improvements will be possible that increase the scope of this type of analysis, including investigations of even older material. However, exceptional longevity of DNA has only been reported from very cool environments (such as those in Greenland). The currently oldest hominin DNA sequences are much younger than 2 million years and were retrieved from the ~430k years old skeletal remains of early Neandertals found in Sima de los Huesos in the Atapuerca Mountains in Northern Spain, an unusually deep cave located at high altitude. Unless very old osseous artefacts are recovered from the permafrost, citing a paper describing the recovery of 2-million-year old DNA from sediment may be seen as a suggestion that our work would bring the field closer to analysing human DNA from early hominins for which there is currently no basis.

H. Clarity and context

The abstract is clear and appropriate, as are the introduction and conclusions. The authors could just emphasize a bit the interest of such studies in places and periods where two populations have cohabited in order to better understand their interactions.

We are glad to hear that the manuscript is well readable also for a non-geneticist. We fully agree that questions about the nature of interactions between different populations, for example during the MP/UP transition, are of central importance for understanding human behaviour in the past. We ultimately hope that the analysis of DNA from artefacts may contribute to addressing some of these questions, as implied in the final sentence of the manuscript: "[...] it might become possible to systematically combine genetic and cultural analyses to study Pleistocene artefact use and uncover possible task specialisation by individuals of a particular biological sex or genetic ancestry."

Prof. Sophie A. de Beaune

3. Referee #3 (Remarks to the Author):

A. The key results of the manuscript consist of the non-destructive extraction of ancient human DNA from a deer pendant found in layer 11 of the Denisova Cave, with a genetic age estimate of 20.2 (range 6.1-33.4 ka) and an attribution to a female individual, genetically close to other Paleolithic human remains, dated to ca. 24 ka and 17 ka, also found in Siberia but further East. Other key findings are the comparisons between the age estimates obtained via the ancient human genetic age attribution of

the pendant, the ancient deer genetic age attribution of the pendant and the radiocarbon dates on other objects found close by in the same archaeological layer.

B. The contribution is original, as it presents for the first-time ancient DNA recovered from a Paleolithic artefact, a perforated deer tooth, i.e. an item known to be used as personal body/garment decoration. Identifying biological sex and genetic ancestry of such an object is significant for the discussions on the origin(s) of personal adornment, symbolism, the trajectories of cultural traditions and on the identification of sex specific material culture in prehistoric societies.

An important element of the contribution is also the elaboration of a non-destructive method for DNA extraction, which as such is not new (cf. references provided by the authors) but very welcome for archaeological research questions addressing rare and unique samples.

Furthermore, the comparison of genetic age estimates from human and faunal aDNA recovered from the same object is highly interesting, even if the range of these estimates is huge and dependent on a number of assumptions regarding population sizes and reproduction rates. The comparison with age estimates obtained via radiocarbon dating is only indirect as it does not apply to the same object, but to objects found close by (a direct radiocarbon date on the deer pendant itself would have been most interesting as well). Those indirect radiocarbon ages are quite different in age from each other. The implications of these discrepancies are not fully discussed.

First, we would like to thank the reviewer for the positive feedback and suggestions.

The two radiocarbon dates obtained from the charcoal samples indicate a relatively large time range for the formation of layer 11 in South Chamber (24-39 ka), not an actual discrepancy between the dates. This was poorly phrased in the manuscript and has now been corrected. Also, since only two radiocarbon dates were obtained, the true time range for the formation of the layer may be even wider.

The confidence intervals on the genetic dates are indeed wide, largely due to the relatively small size of the mitochondrial genome. Importantly, the genetic age estimates for the deer and human mtDNA were obtained independently from each other, yet the point estimates are in good agreement with one another and fall close to one of the two radiocarbon dates obtained from charcoal. The close affinity of the human nuclear DNA to Mal'ta 1 (24 ka) and AG 3 (17 ka) provides a further independent line of evidence supporting a date around 19-25 ka. In light of the consistency of these results, we came to the conclusion that it is more important to preserve the integrity of the object rather than further narrowing down its age by removing a large part of it for radiocarbon dating.

C/F. The justification and followed procedure for establishing the non-destructive DNA isolation method are explicit but not 100% convincing and could benefit from some more details/explanations.

Of the 4 tested reagents, two were excluded because of their possible impact on the surface topography of the sample and one was retained because it offers the possibility of temperature-controlled DNA release. The usefulness of temperature-controlled DNA release is however not explained and also the reason to exclude the fourth reagent is not given.

We agree that an important rationale for the choice of phosphate-based non-destructive DNA extraction was missing in the text. We therefore added an explanation to the methods section: "Step-wise extraction of DNA makes it possible to closely monitor the release of different DNA components during the extraction process (endogenous DNA, environmental DNA from the surrounding sediment, ancient human DNA and present-day contamination), potentially allowing inferences to be drawn

about whether these components originate from traces of sediment that may still be adherent to the object, from its surface or its interior.”.

We have also slightly expanded a sentence in the methods section to make it even clearer why bleach was included in the experiments: “Bleach treatment destroys surface-bound DNA and is not suitable for DNA extraction. However, it may be used in later implementations of the method to remove contaminant DNA prior to non-destructive DNA extraction.” To which extent bleach pre-treatment and phosphate-based non-destructive DNA extraction can be combined is something we are planning to investigate in future work.

Furthermore, the testing of the degree of surface modifications induced by the 4 reagents is not straight forward as the different reagents were applied to different types of raw material (e.g. mammoth ivory + herbivore tooth for GuSCN while carnivore canine + rabbit/hare long bone for EDTA), in addition from different stratigraphic layers and archaeological sites, and for which the comparative states of preservation (e.g. altered or well preserved) and types of surfaces (e.g. smooth or porous, encrusted with adhering exogenous material or not) are not given and for which the initial surface conditions may be different. A testing of the impact of different reagents on identical modern samples and on bones or tooth of the same species from a single chronologically and spatially homogeneous Paleolithic layer and previously cleaned (e.g. in an ultrasonic bath) would have been more convincing.

We agree that it would have been desirable to test the four reagents on a more homogenous set of Pleistocene bones and teeth. Unfortunately, objects that are similar in size and shape to the ones typically used for osseous artefact production were relatively rare in the collections we had permission to use in experiments with a potentially destructive outcome, which led to variability in the test set. We tried to compensate for some of this variability by choosing two objects per condition, preferably a bone and a tooth. We initially considered using modern samples, but were concerned that they would be more resistant to surface alterations and would provide a poor proxy for ancient material in their response to chemical treatment.

It is difficult unfortunately to find terms or parameters to properly characterise or quantitatively measure the state of preservation of old bones. Nonetheless, we have added some descriptive information about the state of preservation of the freshly excavated objects to the supplement, which are most critical in this study due to their reduced load of surface contamination, which hampered the analysis of the other objects.

The non-destructiveness of the method is important if the research question requires an extensive number of samples or if destructive methods would cause a non-acceptable loss of information. If the research question is to identify the best way to extract DNA of the maker/user of a Paleolithic pendant, a destructive micro-sampling of a Paleolithic pendant, repeatedly done in the past for radiocarbon dating, seems justifiable especially given the fact that it is only partially destructive and that methods exist to document the outer topography and inner structure of such objects prior to destruction.

As many factors contribute to DNA preservation and extraction (e.g. Latham KE, Miller JJ. Forensic Sci Res. 2018) the non-destructiveness of the method seems subordinate and feasibility/reliability tests of different methods should primarily be concerned with accuracy and alternative non-destructive methods could subsequently be compared.

Apart from the non-destructiveness, the method we used offers additional advantages. First, by submerging the whole specimen in the extraction buffer, no assumption needs to be made about where specifically in/on the object the human DNA is concentrated. Second, gradual DNA release provides DNA release curves, which are important for demonstrating that the DNA comes from the

object itself rather than from sediment particles that may be adhering to its surface. But we will consider a comparison between non-destructive and destructive sampling in follow-up work.

The relevance of the application of the chosen non-destructive DNA isolation method to a series of different sorts of bones (modified and non-modified, well and badly preserved, different raw materials) from ancient excavations curated for a long time is not clear. These show evidence of contaminations, a fact well known already.

Cleanly excavated osseous artefacts were difficult to obtain, as they were not previously known to be a potential source of ancient human DNA (it took us years to collect four such objects for this study). We therefore first applied the method to material from existing collections. We were hoping that most of the contaminant DNA would be released in the first phosphate fractions and that ancient human DNA from the wearers/users of the objects would be released in subsequent incubation steps. Unfortunately, the contamination problem could not be overcome with this method, in line with the reviewer's expectation. We are planning to test other decontamination procedures in the future, for example bleach treatment, in hopes that this research can be expanded to material from existing collections, which is more readily available.

The conclusion that extended incubation at high temperature best enables the extraction of ancient human DNA could be reinforced if tests were performed for each temperature on the same number of incubations (if understood correctly, the 90°C incubations already passed three incubations for each of the lower temperature settings so a total of 9 previous incubations while the lower temperature settings passed much less incubations).

This is a possibility we will consider in future work if larger numbers of freshly excavated objects become available. In our initial experiments we aimed at producing detailed DNA release curves for each individual object, as we did not know at which temperature the different DNA components would be released.

G/H. The abstract, introduction and conclusions seem appropriate.

The main text and figures however often miss reference to the information and data given in the Supplementary materials which should be mentioned more appropriately.

We have added additional references to the Supplementary Information in the main text. In addition, we made sure that all sections of the Supplementary Information are referenced in the Methods section.

Title reflects only part, albeit the most salient aspect, of the research presented in the paper

We take this as approval. Nonetheless, we came to realise that the title may be too specific with regards to the role of the individual who made or wore the artefact. Given its nature, a pendant, it is most parsimonious to assume that the DNA was deposited by the person wearing it. However, we cannot exclude with certainty that the DNA we isolated may belong to its producer (possibly the same person), especially if it was not worn for a long time. We indicated this possibility throughout the text but not in the title. Thus, for the sake of clarity, we are now suggesting a new title which avoids this distinction: "Ancient human DNA recovered from a Palaeolithic pendant".

Affiliations of the authors: Numbers 6 and 12 refer to the same institution`

Fixed. Thanks for spotting this.

Line 63: the e.g. mentioned in round brackets should come after "body fluids" (not after "porosity")

Fixed. Thanks for spotting the misplacement.

Reference 12 is incomplete.

Fixed. Thanks for spotting this.

Extended data table 1: The description of the samples seems minimalistic. Reference to Supplementary Information SI1 (giving some information of the stratigraphic provenance/dating of the samples) should be made here and also in the main text (Line 80; this would also resolve the problem that SI2 is cited before SI1 in the main text); also, the abbreviations used in the "Reagent tested" column should be explained in the legend.

We agree that stratigraphic and age information is helpful and have modified Extended Data Table 1 accordingly (now also containing the contents of the previous Extended Data Table 2). We have also added a reference to Supplementary information 1 in the main text, right behind the reference to Extended Data Table 1, as well as a footnote to Extended Data Table 1 explaining the abbreviations used for the reagents.

Extended data figure 1: the figures cuts (bottom right) the explanation of the colors (red and yellow) used.

Fixed. Thanks a lot for spotting this.

Supplementary Information: it is a bit confusing that the lines are numbered per page and not as one sequence throughout the whole SI document or at least as a sequence/SI.

Changed as suggested.

Supplementary Information SI1: information on the state of preservation of the samples should be systematically given for all samples, as this variable may be important for the interpretation of the results.

As indicated above, we have now added information on the state of preservation of the freshly excavated artefacts to Supplementary Information 1.

Also, the Headings on page 4 ("Stratigraphic context", "Faunal remains") and page 5 ("Radiocarbon chronology") could mention the archaeological site at stake as it is not immediately clear to which of the multiple samples described in SI1 they refer.

Thanks for this suggestion. To make navigation within the supplement easier we adjusted the formatting of the subheadings.

In the section "A freshly excavated pendant from Denisova Cave": does "in the roof of layer 11" (Line 4) mean "in the upper part of layer 11" (idem Line 13: meaning of "in the roof"?).

This is correct. Changed as suggested.

In the "Radiocarbon chronology" section: it should be specified which date comes from the middle and which from the upper part of layer 11 (does the younger date come from the upper part and the much older date from the lower part of layer 11?

And in what position within layer 11 was the pendant found?).

We rephrased the respective section of SI1 to remove ambiguities about the relative position of the charcoal samples and the pendant to each other. Layer 11 is tilted, but viewed in the sequence of deposition, the old radiocarbon date comes from the lower part of layer 11, the younger radiocarbon date from the middle part and the pendant from the upper part.

Extended data table 2: taxonomic identification would be welcome here as it is directly relevant for the interpretation of the DNA results. Also reference to SI1 seems necessary here as it is not clear to what the abbreviations in the “context information” column refer.

Extended data table 2 was merged with extended data table 1. We agree with reviewer #1 and #3 and adjusted the table accordingly by adding information about the morphological identification. In addition we added a reference to Supplementary information 1 to the main text, right behind the reference to Extended Data Table 1.

Extended data table 3: where do the “311 present-day humans” data come from?

Thanks for pointing this out. We added a reference to the part of the SI that contains information about the 311 present-day humans to the Methods section.

Figure 2: the legend states the figure is concerned with 6 artefacts but the figure provides reference to 5 samples only (DCP1 is represented twice; the reason why is not explained or immediately clear).

This was a copy paste error in the sample names. Thanks for spotting it!

Extended data figure 4: the photo does not really give information on the location of the pendant in the Denisova Cave, rather it gives a sight of the sediment surrounding it. The legend should be adapted or spatial and stratigraphic context information should be given in the figure.

We have now changed the legend of figure 4 to “Photograph of DCP1 as it became exposed during excavation”.

Figure 3: The sources of the human DNA data are not mentioned here. Reference should be made to SI5 and SI7.

We added references to Supplementary information 5 and 7 to the parts of the main text that refer to the respective panels in Figure 3. In addition, we made sure that Supplementary information 5 and 7, which contain full information about the human DNA data used, are also cited in the Methods section.

Supplementary table 3.1: the reason why there is no date provided for most of the samples in this table labelled “Radiocarbon dated samples” is not clear.

We agree that the title of Supplementary table 3.1 was confusing. We changed the title and rephrased the legend to make clear that both collagen yields (for all samples) and radiocarbon dates (for some samples) were obtained to detect possible effects of the non-destructive DNA extraction method (in the presence and absence of Tween-20 in the extraction buffer) on the success of collagen extraction and the accuracy of the dates.

Reviewer Reports on the First Revision:

Referee #2 (Remarks to the Author):

The authors have answered my questions well and I consider their manuscript to be quite suitable for publication as is.

Referee #4 (Remarks to the Author):

This is a very interesting work because it provides evidence to directly link archaeological bone artefacts to their "owners" or "makers"; it is a novel approach and a new, rather unexpected, source of genetic evidence to understand the past that opens new possibilities. I think this could be even more interesting if applied to potential ancient jewelry such as those from Grotte du Renne (Arcy sur Cure) attributed to Neanderthals and published some years ago. This approach could really solve if Neanderthals wore those objects by providing direct evidence. Maybe this potential application could be mentioned also in the manuscript as future applications.

A crucial point in supporting authenticity of any ground-breaking ancient DNA study is the pattern of miscoding lesions at the end of the DNA reads (also the fragment length distribution, maybe they could show this also); this is mentioned in some places of the main text (for instance, in lines 149-150, "Notably, significant signals of cytosine deamination were observed among the human mtDNA fragments recovered in one of the 90°C fractions from DCP1"). The word "significant" is vague here because it is possible to give a quantitative figure (these are shown in Suppl Table 5.1 and they indeed quite high). I would suggest to include the actual number between brackets when appropriate to strengthen the point.

A clear possibility would be that the human DNA detected in this pendant derives from sweating. There is abundant literature on bacteria involved in human skin (also to odour in sweating), including genera *Corynebacterium*, *Staphylococcus* or *Cutibacterium*. I was wondering if the authors could do a metagenomic search on these or other potentially interesting bacteria, taking advantage also of the potentially high damage patterns at the read's ends.

In two places in the text it is mentioned that archaeologists can contaminate remains by handling during excavation: in the Conclusions and also lines 118-119: "Since present-day human DNA contamination is ubiquitous on surfaces of objects that were handled with bare hands during and after excavation." I would suggest to include a reference here.

Author Rebuttals to First Revision:

This is a very interesting work because it provides evidence to directly link archaeological bone artefacts to their "owners" or "makers"; it is a novel approach and a new, rather unexpected, source of genetic evidence to understand the past that opens new possibilities. I think this could be even more interesting if applied to potential ancient jewelry such as those from Grotte du Renne (Arcy sur Cure) attributed to Neanderthals and published some years ago. This approach could really solve if Neanderthals wore those objects by providing direct evidence. Maybe this potential application could be mentioned also in the manuscript as future applications.

First, we thank the reviewer for the positive feedback and suggestions.

As it stands, the method only seems suitable for material excavated under clean conditions (e.g. with gloves). We therefore hesitate to propose its application to collections of previously excavated items, such as the jewelry from Grotte du Renne. As indicated in the concluding paragraph, we hope that our paper will raise awareness for the need of clean excavation protocols, which will ultimately lay the basis for the discovery of new osseous artefacts suitable for identifying the DNA of their users or makers. At which sites this will happen and which specific questions this may help addressing is not in our hands. Therefore, our outlook on future applications remains rather general, but covers questions such as the one raised by the reviewer: "If this is done, it might become possible to systematically combine genetic and cultural analyses to study Pleistocene artefact use and uncover possible task specialization by individuals of a particular biological sex or genetic ancestry."

A crucial point in supporting authenticity of any ground-breaking ancient DNA study is the pattern of miscoding lesions at the end of the DNA reads (also the fragment length distribution, maybe they could show this also); this is mentioned in some places of the main text (for instance, in lines 149-150, "Notably, significant signals of cytosine deamination were observed among the human mtDNA fragments recovered in one of the 90°C fractions from DCP1"). The word "significant" is vague here because it is possible to give a quantitative figure (these are shown in Suppl Table 5.1 and they indeed quite high). I would suggest to include the actual number between brackets when appropriate to strengthen the point.

We agree with the reviewer that the phrasing „significant signals of cytosine deamination“ was too imprecise. The thresholds used to determine significance were previously hidden in SI4 ("The presence of ancient DNA was determined individually for the sequences from each family (and each library) by computing the frequency of terminal C-to-T substitutions as a proxy for the deamination rate at the molecule ends. Substitution frequencies significantly higher than 10% on both molecule ends were then taken as evidence for the presence of ancient DNA from the respective family."). We now moved these sentences to the methods section and added an explanation on how the significance was established ("based on 95% binomial confidence intervals"). This should make it easier for the readers to evaluate the strength of the deamination signals required for authenticating ancient DNA sequences, while avoiding lengthy additions to the main text detailing the deamination rates and their associated confidence intervals.

Following the reviewer's suggestion, we have now added the average fragment lengths of the human mitochondrial and nuclear DNA fragments to Data File S1 and S2.

A clear possibility would be that the human DNA detected in this pendant derives from sweating. There is abundant literature on bacteria involved in human skin (also to odour in sweating), including genera *Corynebacterium*, *Staphylococcus* or *Cutibacterium*. I was wondering if the authors could do a metagenomic search on these or other potentially interesting bacteria, taking advantage also of the potentially high damage patterns at the read's ends.

This is an excellent suggestion and should be part of a follow-up study specifically characterizing the source of ancient human DNA isolated from artefacts (we have also considered analyzing DNA methylation patterns for this purpose). Most of the sequence data generated for this study was obtained by hybridization capture, with only a limited amount of shotgun data being available from one of the highest temperature phosphate fractions recovered from the Denisova Cave pendant. To properly control for example for the absence of soil *Corynebacteria*, it is important to produce shotgun sequence data also for the surrounding sediment and possibly the lower temperature phosphate fractions. This is beyond the scope of the present work, which highlights ancient artefacts as a new source of ancient DNA and focuses on the implications of this finding for linking cultural and genetic records in the Pleistocene.

In two places in the text it is mentioned that archaeologists can contaminate remains by handling during excavation: in the Conclusions and also lines 118-119: "Since present-day human DNA contamination appears to be ubiquitous on surfaces of objects that were handled with bare hands during and after excavation." I would suggest to include a reference here.

The statement quoted by the reviewer was not intended as a generalizing statement, but follows directly from the results of our experiments in the previous section. To make this clearer we modified the sentence as follows: "Since present-day human DNA contamination seemed to be ubiquitous on surfaces of objects that were handled with bare hands during and after excavation, [...]". Likewise, the notion in Conclusions that "[...] surface DNA contamination can hamper these analyses [...]" derives directly from the experimental evidence provided in the current work. Although there is plenty of anecdotal evidence in literature for surface contamination of ancient bones and teeth, we are not aware of a study that investigated this problem fully systematically and that could serve as a better reference than the results we present here.